# SORLA regulates endosomal trafficking and oncogenic fitness of HER2

Mika Pietilä[1,7], Pranshu Sahgal[1,7], Emilia Peuhu [1], Niklas Z. Jäntti[1], Ilkka Paatero [1], Elisa Närvä[1], Hussein Al-Akhrass[1], Johanna Lilja[1], Maria Georgiadou[1], Olav M. Andersen[2], Artur Padzik[1], Harri Sihto [3], Heikki Joensuu[3], Matias Blomqvist[4], Irena Saarinen[4], Peter J. Boström[5], Pekka Taimen[4] & Johanna Ivaska [1,6]

The human epidermal growth factor receptor 2 (HER2) is an oncogene targeted by several kinase inhibitors and therapeutic antibodies. While the endosomal trafficking of many other receptor tyrosine kinases is known to regulate their oncogenic signalling, the prevailing view on HER2 is that this receptor is predominantly retained on the cell surface. Here, we find that sortilin-related receptor 1 (SORLA; *SORL1*) co-precipitates with HER2 in cancer cells and regulates HER2 subcellular distribution by promoting recycling of the endosomal receptor back to the plasma membrane. SORLA protein levels in cancer cell lines and bladder cancers correlates with HER2 levels. Depletion of SORLA triggers HER2 targeting to late endosomal/lysosomal compartments and impairs HER2-driven signalling and in vivo tumour growth. SORLA silencing also disrupts normal lysosome function and sensitizes anti-HER2 therapy sensitive and resistant cancer cells to lysosome-targeting cationic amphiphilic drugs. These findings reveal potentially important SORLA-dependent endosomal trafficking-linked vulnerabilities in HER2-driven cancers.

[1] Turku Bioscience Centre, University of Turku and Åbo Akademi University, FI-20520 Turku, Finland. [2] Danish Research Institute of Translational Neuroscience Nordic-EMBL Partnership (DANDRITE), Department of Biomedicine, Aarhus University, Ole Worms Allé 3, 8000 Aarhus, Denmark. [3] Laboratory of Molecular Oncology, Translational Cancer Biology Program, University of Helsinki and Comprehensive Cancer Center, Helsinki University Hospital, FI-00290 Helsinki, Finland. [4] Institute of Biomedicine, University of Turku and Department of Pathology, Turku University Hospital, FI-20520 Turku, Finland. [5] Department of Urology, University of Turku and Turku University Hospital, FI-20520 Turku, Finland. [6] Department of Biochemistry, University of Turku, FI-20520 Turku, Finland. [7] These authors contributed equally: Mika Pietilä, Pranshu Sahgal. Correspondence and requests for materials should be addressed to M.P. (email: mika.pietila@utu.fi) or to J.I. (email: Johanna.ivaska@utu.fi)

The human epidermal growth factor receptor 2 (HER2; also known as ErbB2) is a receptor tyrosine kinase and a well-established oncogene. HER2 amplification is found in 15–20% of breast cancers[1,2], and HER2 overexpression or activating mutations are clinically relevant in other solid tumours such as bladder cancer, gastric, colorectal, and lung adenocarcinoma[3–5]. The biological relevance of HER2 as a driver oncogene is undisputed, and several targeted therapies have been approved for treating HER2-dependent cancers.

Several previous studies, including ours, demonstrate that the oncogenic signalling and the endosomal traffic of many receptor tyrosine kinases are functionally coupled[6–9]. Endosomal trafficking critically controls, for instance, the strength and duration of signals emanating from the epidermal growth factor receptor (EGFR; also known as ErbB1) and MET[6,10,11]. However, in comparison to other receptor tyrosine kinases, the details of HER2 trafficking are poorly understood. The prevailing view, supported by two different models, is that HER2 resides almost exclusively on the plasma membrane in HER2-amplified cancer cell lines[12]. Data in favour of a 'limited internalization' model suggest that HER2 resists internalization due to (1) the absence of identified ligands required for induction of endocytosis[13], (2) the absence of a recognizable internalization motif[14], (3) inhibition of clathrin-coated pit formation[15,16], (4) association with membrane protrusions[17] and (5) heat shock protein 90 (HSP90)-dependent stabilization of HER2 on the plasma membrane[18–20]. On the other hand, data in support of a model for 'rapid recycling' resulting in nearly exclusive cell-surface localization of HER2 have been brought forward[21–23]. Hence, key outstanding questions are: does HER2 undergo endosomal trafficking in HER2-driven cancer cells, and what would be the functional consequence of HER2 trafficking for its oncogenic properties?

SORLA is a multifunctional intracellular sorting protein belonging to the sortilin and LDL-receptor families consisting of a large extracellular domain, a transmembrane domain and a short cytoplasmic tail[24], which is essential for its membrane sorting functions[25,26]. SORLA has been implicated in regulating amyloid precursor protein (APP) processing during the pathogenesis of Alzheimer's disease (AD)[27–29] and in lipid metabolism and obesity[30–33]. We made the unexpected observation that SORLA is highly expressed specifically in HER2-driven cancers where SORLA levels correlate with HER2 subcellular localization. We find that SORLA co-precipitates with HER2 and regulates its trafficking from intracellular compartments to the plasma membrane. However, at this point, we do not know whether SORLA interacts directly with HER2 or if they are coupled through a molecular complex or an intermediary partner. Regardless, we show that SORLA is necessary for oncogenic HER2 signalling in vitro and in vivo. Depletion of SORLA induces lysosome dysfunction and results in mislocalization of signalling-defective HER2 protein to lysosomes. As a result, SORLA depletion halts tumorigenesis and sensitizes cancer cells to a clinically relevant lysosome-targeting drug. Taken together, our data demonstrate that SORLA has previously unrecognized oncogenic functions in carcinomas.

## Results

### SORLA co-localizes with HER2 on the plasma membrane and in intracellular vesicles

Analysis of different breast cancer cell lines revealed prevalent SORLA protein expression in cells with HER2-amplification (Fig. 1a). In addition, SORLA was highly expressed in the 5637 bladder cancer cell line harbouring a HER2-activating mutation (S310F)[34] when compared to the HER2-low T24 cell line and a primary patient-derived bladder cancer cell line (Supplementary Fig. 1a). As SORLA has not previously been scrutinized in carcinoma cells, we first examined cellular SORLA levels in a quantitative manner. Flow cytometry (FACS) analysis revealed BT474 cells to have the highest cell-surface levels of SORLA compared to MDA-MB-361 cells (intermediate expression) and JIMT-1 cells (lowest expression) (Fig. 1b), further validating the correlation between total SORLA and HER2 levels observed by western blotting (Fig. 1b; Supplementary Fig. 1b). Next, we investigated the subcellular localization of SORLA in different endosomal compartments. Endogenous SORLA was found to localize largely to early endosomes (identified by EEA1 and Rab5 expression) and retrograde vesicles (VPS35) in MDA-MB-361 breast cancer cells (Supplementary Fig. 1c). The SORLA-GFP fusion protein was similarly detected in EEA1- and VPS35-positive vesicles, and while present in recycling endosomes (Rab11) (Supplementary Fig. 1c), was not detected in late endosomes (Rab7) or lysosomes (LAMP1) (Supplementary Fig. 1c).

Previous studies have indicated that HER2 is mainly restricted to the plasma membrane[12]; however, these observations were based on a very limited number of cell lines and/or on exogenous HER2 overexpression[13,17,21]. Since SORLA, an important sorting protein in neuronal cells and adipocytes, is expressed in HER2-positive cancer cells (Fig. 1a), we were interested to re-examine the subcellular localization of endogenous HER2 in breast cancer cells with variable levels of SORLA expression (Fig. 1a, c; Supplementary Fig. 1d). In the BT474 and SKBR3 cells with high SORLA expression, HER2 was mainly on the plasma membrane, in accordance with previous work[17,21], and did not overlap with EEA1-positive endosomes (Fig. 1a, c, Supplementary Fig. 1d). A similar distribution was apparent in the third SORLA-high cell line HCC1419 (Fig. 1a, Supplementary Fig. 1d). However, in cells with intermediate (MDA-MB-361) or low (HCC1954 and JIMT-1) SORLA expression, HER2 was also distributed intracellularly indicating that HER2 must be trafficked at least in a subset of HER2 cancers (Fig. 1c, Supplementary Fig. 1d). The distinct cell-surface HER2 levels were further validated with FACS (Supplementary Fig. 1e). With respect to the intracellular HER2 pool, a proportion of the receptor demonstrated a clear overlap with EEA1, indicating localization in early endosomes (Fig. 1c, d). In MDA-MB-361 cells, endogenous SORLA co-localized with HER2 in EEA1-positive early endosomes (Fig. 1d), and in the SORLA-low JIMT-1 cells, SORLA-GFP localized to HER2 and VPS35-positive intracellular vesicles (Fig. 1d). When SORLA-GFP expression was analysed in JIMT-1 cells ($n = 130$), $19.8 \pm 0.8$ percent of SORLA-GFP overlapped with intracellular HER2 (Supplementary Fig. 1f) showing Pearson's $R$ of $0.3 \pm 0.01$ indicative of partial co-localisation.

To study whether SORLA and HER2 would show similar dynamics in cells, we chose to image the MDA-MB-361 cells expressing intracellular as well as cell-surface pools of HER2. For visualization, we performed live-cell TIRF imaging (allowing visualization of events close to the plasma membrane) of SORLA-GFP and HER2 labelled with Alexa568-conjugated anti-HER2 antibody (trastuzumab; Tz-568). Short-lived SORLA- and HER2-positive structures were detected in the TIRF-plane, indicative of active dynamics to and from the plasma membrane. In addition, co-localizing puncta of SORLA and HER2 were frequently observed undergoing dynamic lateral movement on the plasma membrane (Supplementary Fig. 1g and Supplementary Movie 1). Live-cell imaging deeper in the cytoplasm showed that SORLA and HER2 move together within the same endosomal structures (Supplementary Fig. 1g and Supplementary Movie 2). Collectively, these data demonstrate that SORLA and HER2 undergo co-trafficking between the plasma membrane and endosomes.

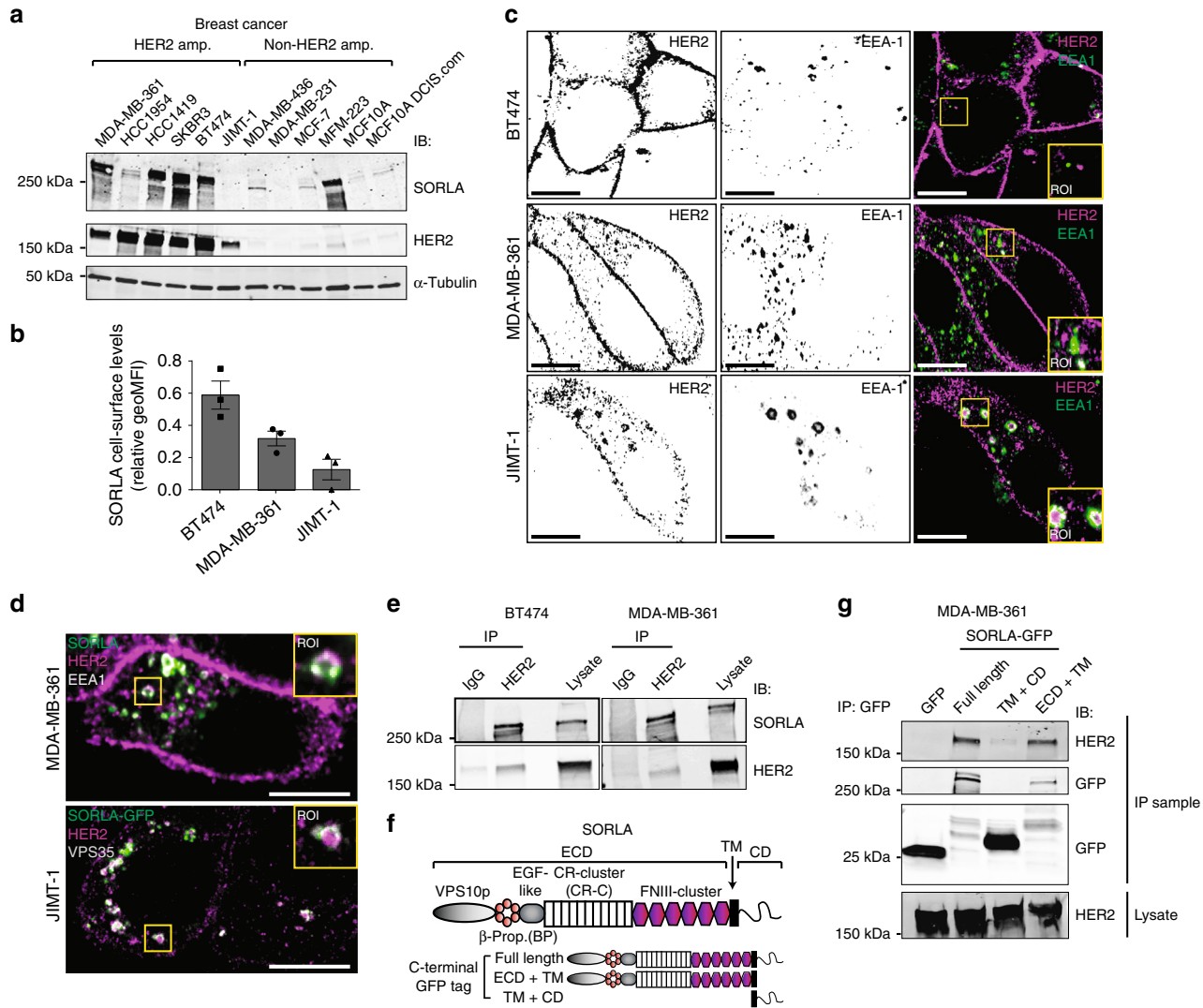

**Fig. 1** SORLA is highly expressed in HER2-amplified breast cancer cells and co-traffics with HER2. **a** Western blot analysis of SORLA and HER2 protein levels in breast cancer cell lines. α-tubulin is a loading control. **b** Quantification of SORLA cell-surface levels by FACS in MDA-MB-361, BT474 and JIMT-1 cells ($n = 3$ independent experiments; data are geo mean fluorescence intensity (MFI) ± standard error of mean (s.e.m.). **c** Confocal microscopy imaging of HER2 (magenta) and EEA-1 (green) in BT474, MDA-MB-361 and JIMT-1 cells. Co-localisation of the HER2 and EEA1 signals is indicated in white in the merged panels. **d** Endogenous SORLA (green), HER2 (magenta) and EEA-1 (white) staining in MDA-MB-361 cells (top panel). Endogenous HER2 (magenta) and VPS35 (white) staining in JIMT-1 cells expressing SORLA-GFP (green) (bottom panel). **e** Co-immunoprecipitation of endogenous SORLA with endogenous HER2 in MDA-MB-361 and BT474 cells. **f** Schematic of the SORLA protein domains and summary of the constructs used. **g** Co-immunoprecipitation of endogenous HER2 with different SORLA-GFP fragments in MDA-MB-361 cells. Scale bars: 10 μm. Where immunoblots and micrographs are shown, these are representative of $n = 3$ independent experiments; IB immunoblotting, IP immunoprecipitation. ECD extracellular domain, TM transmembrane domain, CD cytosolic domain

**The SORLA extracellular domain is required for SORLA–HER2 complex formation**. Intrigued by the apparent co-trafficking of SORLA and HER2, we next performed a set of co-immunoprecipitation assays to investigate whether HER2 and SORLA associate. We found that endogenous HER2 and SORLA co-precipitate in MDA-MB-361 and BT474 cells, indicating that HER2 and SORLA may exist in the same protein complex (Fig. 1e). SORLA consists of an extracellular domain (ECD), a transmembrane domain (TM) and a short cytosolic domain (CD) (Fig. 1f). To dissect the SORLA—HER2 association further, we generated truncated SORLA-GFP fusions consisting of either the SORLA extracellular and transmembrane domains (ECD + TM) or the SORLA transmembrane and cytosolic domains (TM + CD) (Fig. 1f, g). HER2 co-precipitated with the full-length SORLA-GFP and with SORLA-GFP ECD + TM in cells, but

failed to associate with SORLA-GFP TM + CD (Fig. 1g). Interestingly, SORLA-GFP TM + CD showed similar vesicular localization as full-length SORLA-GFP, whereas SORLA-GFP ECD + TM was found diffusely in membrane-compartments in the cytoplasm and on the plasma membrane (Supplementary Fig. 2a). Thus, while the SORLA ECD is necessary for the SORLA-HER2 protein complex, the SORLA CD appears to be required for correct subcellular localization of SORLA.

The SORLA ECD is subdivided into five domains: an N-terminal VPS10p domain followed by a β-propeller (BP), an EGF-like (EGF) domain, a complement type repeat-cluster (CR-C) and a FNIII-domain cluster (Supplementary Fig. 2b). To investigate which domain of SORLA is required for the SORLA—HER2 complex formation, we produced and purified myc and 6xHIS-tagged full-length SORLA ECD, and SORLA ECD fragments

(CR-C, BP-EGF and BP-EGF + CR-C). Pull-down assays with the recombinant fragments showed that the full-length SORLA ECD forms a complex with endogenous HER2 (BT474 cell lysate) (Supplementary Fig. 2c). In fact, all ECD fragments tested pulled down HER2 (Supplementary Fig. 2c), suggesting that several, potentially weak affinity, direct or indirect extracellular interactions regulate the SORLA─HER2 complex formation.

**SORLA regulates HER2 cell-surface levels and HER2 oncogenic signalling**. The apparent inverse correlation between SORLA levels and the proportion of intracellular HER2 in the different HER2 cell lines (Fig. 1a, c, Supplementary Fig. 1d) prompted us to hypothesize that cell-surface HER2 levels may be regulated by SORLA. To test this, we performed loss-of-function experiments in high-SORLA BT474 cells and gain-of-function experiments in intermediate/low SORLA cell lines MDA-MB-361 and JIMT-1 cells, respectively. In BT474 cells, with predominantly plasma membrane-localized HER2 and high SORLA expression, silencing of SORLA resulted in, approximately, a 50% decrease in cell-surface HER2 protein levels (Fig. 2a). Conversely, in the SORLA-intermediate MDA-MB-361 and SORLA-low JIMT-1 cells, in which HER2 localizes more to endosomal structures, SORLA overexpression increased cell-surface HER2 levels significantly (Fig. 2a). Total HER2 protein levels followed a similar trend of being significantly downregulated in SORLA-silenced BT474 cells and upregulated in SORLA-overexpressing MDA-MB-361 and JIMT-1 cells (Fig. 2b, c). Although the reduction in total HER2 protein levels upon SORLA silencing was observed consistently, its extent varied among experiments. Quantitative PCR analysis of *ERBB2* mRNA levels after SORLA silencing or overexpression did not show any significant differences indicating that SORLA-mediated regulation of HER2 occurs predominantly at the post-transcriptional level (Supplementary Fig. 3a). These effects of SORLA silencing may not be limited to regulation of HER2 alone; we find that cell-surface β1-integrin levels were also reduced upon SORLA silencing (Supplementary Fig. 3b).

HER2 amplification is a major driver of proliferation and tumorigenesis, which prompted us to explore whether SORLA plays a functional role in breast cancer cells. Efficient silencing of SORLA in HER2-amplified, SORLA-high BT474 cells significantly reduced cell proliferation (Fig. 2d, Supplementary Fig. 3c −d). This was specifically due to loss of SORLA rather than off-target effects, given that silencing SORLA with five individual siRNAs significantly reduced BT474 cell proliferation (Supplementary Fig. 3d, e). Conversely, in the SORLA-low JIMT-1 cells, expression of SORLA-GFP, triggering HER2 upregulation on the plasma membrane (Fig. 2a), significantly increased proliferation of these cells (Fig. 2e, Supplementary Fig. 3f). When SORLA was silenced in non-HER2-amplified but FGFR2-amplified MFM-223 breast cancer cells (Fig. 1a), there was no effect on cell proliferation (Supplementary Fig. 3g), indicating that SORLA is required for proliferation only in HER2-dependent cancer cells.

HER2 signalling on the plasma membrane along the PI3K/ AKT pathway is critical for HER2 growth-promoting functions in cancer cells. Silencing of SORLA in BT474 and MDA-MB-361 cells led to decreased phosphorylation of AKT (Ser473) and 4E-BP1 (Thr37/46) as well as decreased cyclin D1 levels, but did not inhibit mitogen-activated protein kinase (ERK1/2) signalling (Fig. 2f, Supplementary Fig. 3h). Taken together these data suggest that SORLA silencing specifically attenuates cell proliferation and PI3K-dependent HER2 signalling in HER2-amplified cells.

To investigate further the requirement of SORLA expression for the proliferation of HER2-dependent cancer cells, we silenced SORLA in the intermediate SORLA expressers, MDA-MB-361

cells, with a 3′UTR targeting siRNA and then re-expressed SORLA in the same cells. Re-expression of SORLA-GFP in SORLA-silenced MDA-MB-361 cells fully rescued cell proliferation back to control levels (Fig. 2g, Supplementary Fig. 3i). Importantly, only full-length SORLA-GFP, and not the SORLA fragments lacking either HER2 binding function (SORLA-GFP TM + CD) or correct subcellular localization (ECD + TM), was sufficient to rescue the effect of SORLA silencing on MDA-MB-361 cell proliferation (Fig. 2h). This suggests that the direct or indirect interaction of SORLA with HER2 and SORLA sorting functions are both necessary for SORLA-mediated proliferation of HER2-dependent cancer cells.

Importantly, silencing SORLA compromised the in vivo tumour engraftment of HER2-amplified breast cancer cells in an orthotopic model. SORLA-silenced and control MDA-MB-361 cells were generated by transducing two short hairpin RNAs (shRNAs) targeting SORLA (shSORLA #1 and shSORLA #4) and a non-targeting control (shCTRL). Efficient SORLA silencing strongly inhibited in vitro proliferation of these cells (Supplementary Fig. 3j, k). When control-silenced cells were injected into the mammary ducts of immunocompromised NOD.SCID mice, multiple ductal carcinoma in situ (DCIS) lesions were formed within 10 weeks. In contrast, the development of DCIS tumours from SORLA-silenced xenografts was almost completely halted (Fig. 2i). Taken together these data indicate that SORLA functionally regulates both the expression and the oncogenic function of HER2 in breast cancer.

**SORLA promotes HER2 recycling**. The steady-state distribution of cell-surface receptors between endosomes and the plasma membrane is regulated by the respective rates of receptor internalization and recycling back to the plasma membrane. To study whether SORLA regulates this balance in HER2 cell lines, we investigated HER2 localization and trafficking in the presence and absence of SORLA. First, we silenced SORLA in BT474 cells with high SORLA levels and predominantly plasma membrane HER2 localization at steady state. Interestingly, shRNA-mediated SORLA silencing led to increased intracellular accumulation of HER2, normally not observed in these cells (Fig. 3a, b). A similar shift in HER2 subcellular localization was triggered when inhibiting vesicular recycling with primaquine, indicating that HER2 undergoes constant endocytosis balanced with very rapid recycling in cells with predominantly plasma membrane-localized HER2 and that SORLA may play a role in facilitating HER2 transport (Fig. 3c, d).

Next, we investigated HER2 dynamics in MDA-MB-361 cells, where HER2 is localized both endosomally and on the plasma membrane, and in which SORLA is expressed at intermediate levels. SORLA-silenced MDA-MB-361 cells were subjected to an imaging-based receptor uptake assay. Cell-surface HER2 was labelled with AlexaFluor 568-conjugated trastuzumab (Tz-568) on ice and receptor internalization was then induced by placing cells at +37ºC for 15, 30, and 60 min before fixation. SORLA-silenced cells showed significantly greater accumulation of intracellular HER2 after 30 min of internalization when compared to control-silenced cells (Fig. 3e, f), suggesting a possible defect in recycling. To test this, we labelled cell-surface HER2 as above, allowed receptor trafficking for 45 min, removed the remaining cell-surface-bound Tz-568 antibody with an acid wash and then allowed receptor recycling to occur for 30 min. SORLA-silenced MDA-MB-361 cells demonstrated significantly greater retention of intracellular antibody, indicative of attenuated recycling (Fig. 3g). To investigate the role of SORLA in receptor traffic further, we utilized JIMT-1 cells, which express very low endogenous SORLA, contain a substantial fraction of intracellular

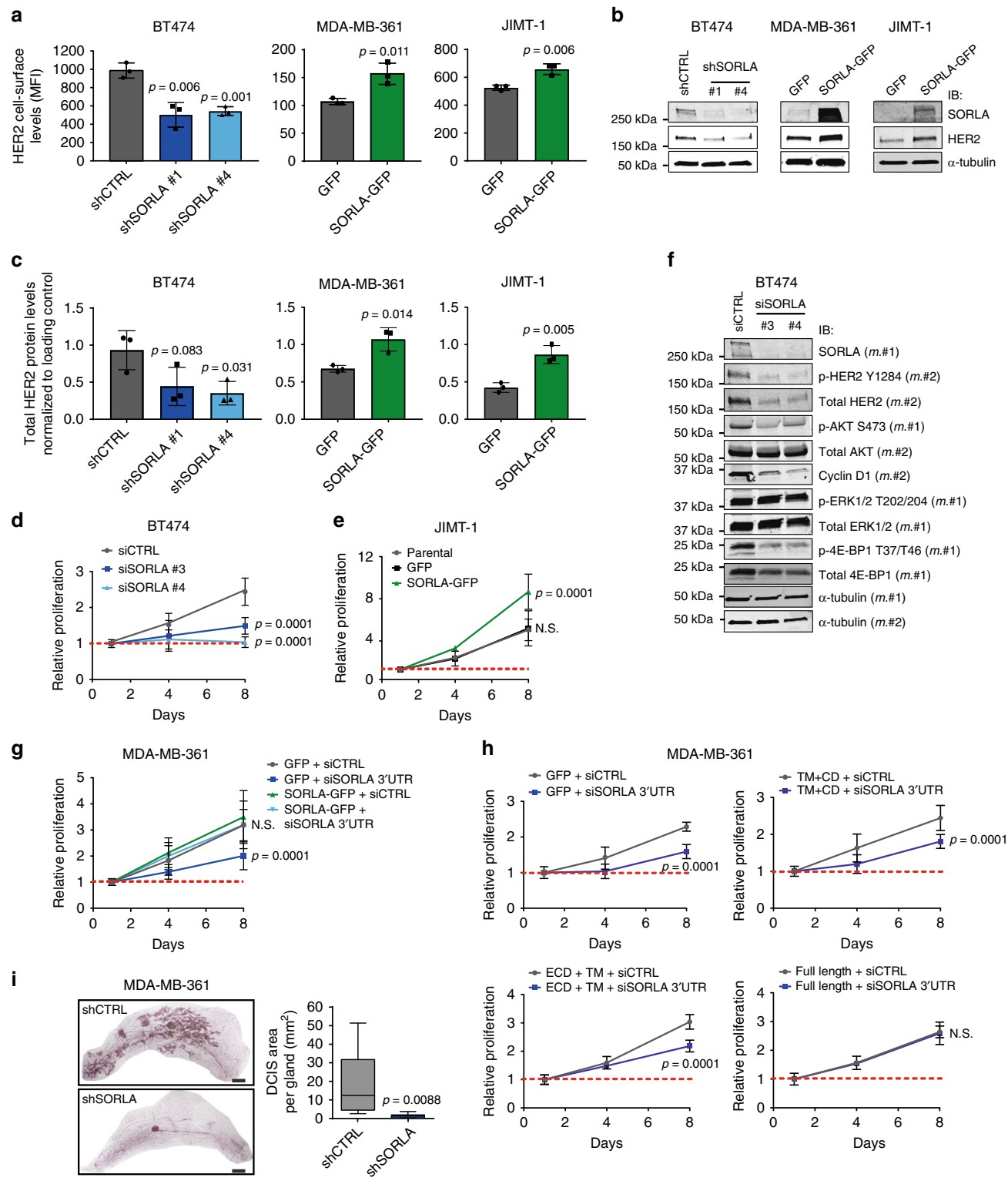

HER2 at steady state and where the overexpression of SORLA leads to increased HER2 cell-surface levels (Fig. 2a). Over-expression of SORLA-GFP in these cells prevented the intracellular accumulation of cell-surface biotinylated HER2 when compared to control transfected cells (Fig. 3h), suggesting possibly accelerated recycling. Furthermore, a biochemical recycling assay, in which biotin is cleaved off from an internalized biotinylated receptor upon recycling back to the plasma membrane, revealed that SORLA-GFP-expressing JIMT-1 cells recycle significantly more HER2 to the plasma membrane compared to JIMT-1 control cells (Fig. 3i). A more detailed time course (5, 10 and 15 min of recycling) indicated that HER2 recycling is higher in SORLA-GFP cells, compared to GFP cells, only at 10 min, possibly due to rapid re-endocytosis of the recycled HER2 in the SORLA-GFP cells (Supplementary Fig. 4a). Interestingly, SORLA silencing also attenuated the recycling of

**Fig. 2** SORLA regulates HER2 cell-surface levels and oncogenic signalling in breast cancer cells. **a−c** SORLA-high BT474 cells were subjected to shRNA-mediated control (shCTRL) or SORLA (shSORLA #1 and #4) silencing. SORLA-intermediate/low MDA-MB-361/JIMT-1 cells were transfected with SORLA-GFP or GFP alone. Flow cytometry analysis of cell-surface HER2 levels (**a**, MFI ± standard deviation (s.d.); $n = 3$ independent experiments), immunoblotting of total HER2 and SORLA, where α-tubulin is a loading control (**b**) and quantification of total HER2 protein levels relative to loading control (**c**) are shown (mean ± s.d.; $n = 3$ independent experiments; statistical analysis: unpaired Student's $t$ test). **d** Proliferation of BT474 cells after SORLA silencing with siRNAs (mean ± s.d. of $n = 12$, four technical replicates, three independent experiments; statistical analysis: two-way ANOVA). **e** Proliferation of parental, GFP control and SORLA-GFP overexpressing JIMT-1 cells (mean ± s.d. of $n = 12$, four technical replicates, three independent experiments; statistical analysis: two-way ANOVA). **f** Western blot analysis of the indicated signalling proteins (phosphorylated and total) in BT474 cells after scramble (siCTRL) and SORLA silencing (siSORLA #3 and siSORLA #4). α-tubulin is a loading control. m.#1 indicates membrane number 1 and m.#2 indicates membrane number 2 ($n = 2$ independent experiments). **g** Proliferation of MDA-MB-361 cells expressing SORLA-GFP or GFP-control after silencing with control siRNA (siCTRL) or siRNA against the 3′UTR of SORLA (siSORLA 3′UTR) (mean ± s.d. of $n = 14$ from four independent experiments, two-way ANOVA). **h** Proliferation of MDA-MB-361 cells expressing SORLA-GFP constructs (full-length, ECD-TM or TM-CD) after silencing with siCTRL or siSORLA 3′UTR (mean ± s.d. of $n = 8$ from two independent experiments; statistical analysis: two-way ANOVA). **i** Quantification of mammary DCIS tumours (Carnoy staining) 10 weeks after injection of shSORLA- and shCTRL-silenced MDA-MB-361 cells into the mammary ducts of NOD.SCID mice (box plot represents median and 25th and 75th percentiles—interquartile range; IQR—and whiskers extend to maximum and minimum values; $n = 9$ mice per group; statistical analysis: unpaired Student's $t$ test). Scale bars: 2 mm

β1-integrins (without influencing endocytosis; Supplementary Fig. 4b, c), suggesting that in cancer cells SORLA could be linked to trafficking of other receptors in addition to HER2. Taken together, these findings demonstrate a role for SORLA in the regulation of HER2 recycling.

**Silencing SORLA triggers HER2 accumulation in dysfunctional lysosomes.** As SORLA silencing triggered increased intracellular retention of HER2 in the internalization assays, we wanted to investigate the subcellular localization of HER2 in SORLA-silenced MDA-MB-361 cells. Imaging revealed striking accumulation of HER2 in enlarged LAMP-1-positive structures (late endosomes/lysosomes), not observed in control cells (Fig. 4a, Supplementary Fig. 5a). The accumulation of HER2 in these structures, within SORLA-silenced cells, is rather surprising as under normal conditions, lysosomal targeting of growth factor receptors is linked to rapid receptor degradation. However, HER2 protein levels in SORLA-silenced cells were only modestly reduced (Supplementary Fig. 5a), suggesting that attenuated HER2 signalling (Fig. 2f, Supplementary Fig. 3i) is not linked to receptor degradation. Instead, since the ligand for PI3K, PI(4,5)P2, is predominantly enriched on the plasma membrane and considered to be absent from late endosomes, this may explain the restricted HER2 signalling, through the PI3K/AKT pathway, from these compartments in SORLA-silenced cells.

Given the enlarged LAMP-1 structures and the apparent discrepancy between increased lysosomal localization of HER2 versus minimal effects on HER2 levels, we investigated whether loss of SORLA could be potentially linked to abnormal lysosome function. Our hypothesis was supported by strong perinuclear accumulation of LAMP1- and CD63 (LAMP3)-positive late endosomes/lysosomes in SORLA-silenced cells compared to control cells (Fig. 4b, c, Supplementary Fig 5b, c). Lysosomal aggregation was confirmed with four different siRNAs targeting SORLA in both MDA-MB-361 and BT474 cells (Supplementary Fig. 5d). Thus, depletion of endogenous SORLA in breast cancer cells leads to altered subcellular localization of lysosomes. Interestingly, this is linked to the altered traffic of HER2, as dual silencing of HER2 and SORLA reduced lysosomal aggregation but did not fully revert the phenotype (Fig. 4d). This suggests that in SORLA-silenced cells, HER2 localization to lysosomes contributes to compromised lysosomal function. This is further supported by the fact that SORLA silencing in the non-HER2-amplified breast cancer cell line MFM-223 does not affect the subcellular localization of lysosomes (Supplementary Fig. 5e).

Further analyses using transmission electron microscopy (TEM) revealed enlarged lysosomes in SORLA-silenced cells

suggesting potential lysosome maturation defects (Fig. 4e). To monitor the proteolytic activity of lysosomes, we analysed loss of quenching of a fluorogenic protease substrate, DQ Red BSA, loaded into cells[35]. SORLA-silenced MDA-MB-361 cells showed significantly lower DQ Red BSA signal (indicating reduced lysosomal cleavage of BSA), detected either by confocal microscopy imaging (Supplementary Fig. 5f) or by flow cytometry, than the respective control cells (Fig. 4f). These data together indicate a link between SORLA-dependent HER2 signalling and lysosome integrity in HER2-driven cancer cells.

**Depletion of SORLA renders HER2-driven cancer cells sensitive to CADs.** Previous studies indicate that cancer cells possess functionally abnormal lysosomes making them more susceptible to cationic amphiphilic drugs (CADs), a heterogeneous class of molecules with a similar chemical structure resulting in lysosomal accumulation and increased lysosomal membrane permeabilization[36]. Recently, cancer cells were shown to be more sensitive to CAD-induced cell death than non-transformed cells[37]. Given the defective lysosomes of SORLA-depleted cells, we wanted to evaluate the response of anti-HER2 therapy-sensitive BT474 and therapy-resistant MDA-MB-361 cells to the antihistamine ebastine, a CAD with cytotoxic effects in lung cancer[37]. Interestingly, ebastine displayed significantly lower IC50 values when tested for growth inhibitory effects in both of these HER2-amplified breast cancer cells following SORLA silencing (Fig. 4g). Moreover, treatment of SORLA-silenced MDA-MB-361 cells with 15 μM ebastine, which is close to the determined IC50 value that inhibits the growth of these cells, significantly increased the levels of cleaved PARP1, indicative of apoptosis, whereas no such effect was seen in control-silenced cells (Fig. 4h, i). Thus, depletion of SORLA increases the sensitivity of breast cancer cells to CADs, regardless of susceptibility to anti-HER2 therapy, presumably due to apoptosis triggered by lysosomal dysfunction. These data indicate that compromised lysosomal integrity downstream of SORLA depletion could be exploited therapeutically to induce cell death in anti-HER2 therapy-resistant breast cancer cells.

**High SORLA expression correlates with poor patient outcome specifically in HER2-amplified breast cancer patients.** Our in vitro data and observations of reduced tumour formation in mice demonstrate an important role for SORLA in regulating HER2 function in breast cancer cells. Next, we evaluated HER2 levels and SORLA expression in clinical specimens of breast cancer. Immunohistochemical staining of SORLA in a breast cancer tissue microarray (TMA) revealed that a substantial

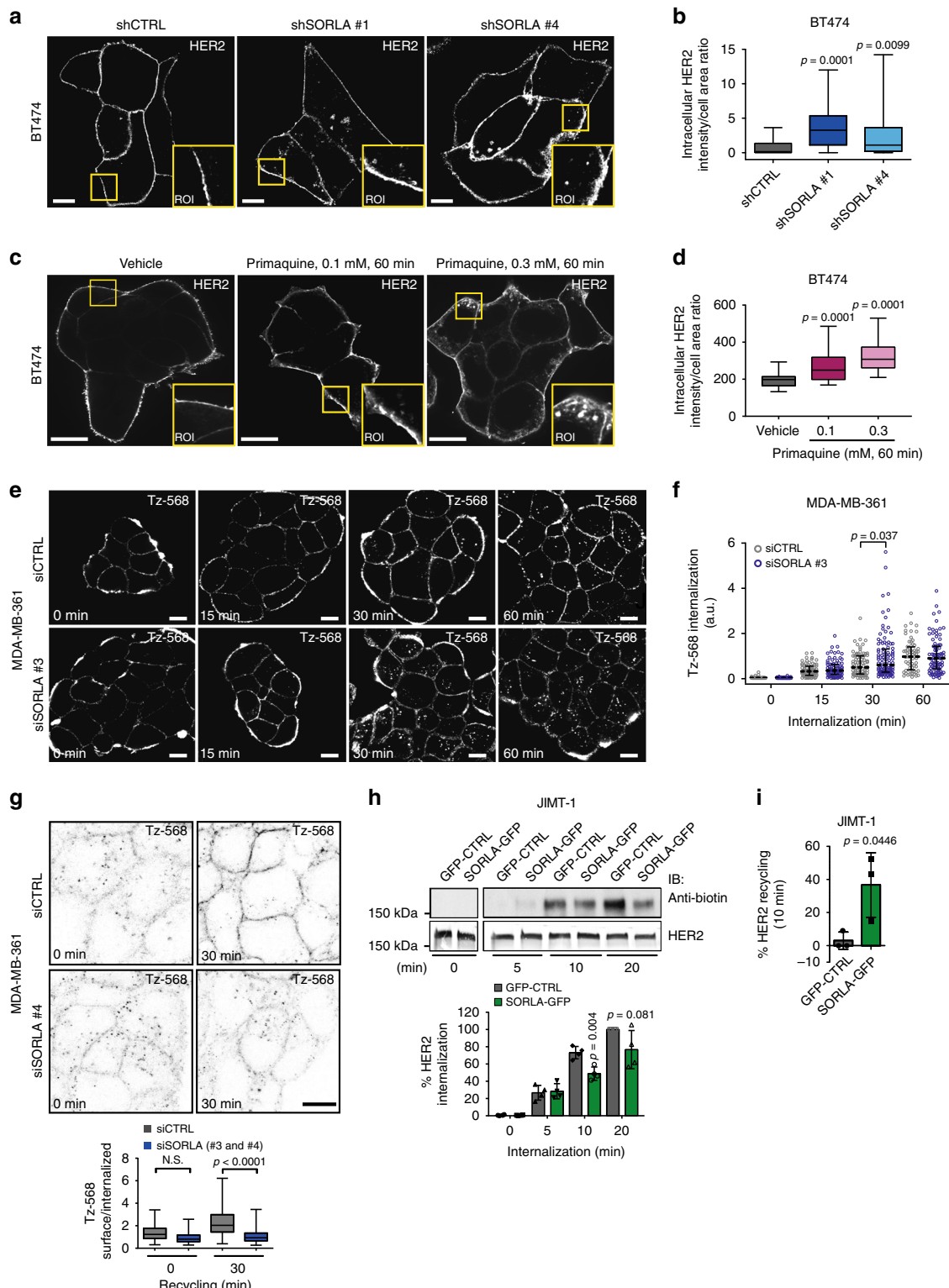

proportion (38%) of HER2-amplified breast cancers express moderate to high SORLA, indicating that HER2-amplified breast cancers fall into two subtypes with respect to SORLA positivity (Fig. 5a). However, in spite of a lack of overall correlation between HER2 amplification and SORLA expression in breast cancer, high *SORL1* expression appears to predict poor relapse-free and overall survival specifically within HER2-amplified breast cancer patients (Fig. 5b) according to the in silico biomarker assessment tool[38] (http://kmplot.com), which interrogates large

datasets (e.g. The Cancer Genome Atlas (TCGA), the European Genome Phenotype Archive (EGA) and the Gene Expression Omnibus (GEO)) in its analysis. Thus, SORLA levels could have a prognostic value in HER2-amplified breast cancers.

**SORLA correlates with HER2 and regulates proliferation and tumour growth in bladder cancer.** To broaden our study for other cancer types in which HER2 overexpression is common and

**Fig. 3** SORLA promotes HER2 recycling. **a**, **b** Confocal microscopy images (**a**) and quantification (**b**) of HER2 staining after SORLA silencing (shSORLA #1 and shSORLA #4) in BT474 cells (n = 31 shCTRL, 25 shSORLA #1 and 20 shSORLA #4 cells from two experiments; analysis performed on 8-bit images; statistical analysis: Mann−Whitney test). **c**, **d** Confocal microscopy images (**c**) and quantification (**d**) of HER2 in vehicle- and primaquine-treated (60 min) BT474 cells (n = 34 (vehicle), 28 (0.1 mM primaquine) and 34 (0.3 mM primaquine) cells; analysis performed on 16-bit images; statistical analysis: Mann−Whitney test). **e**, **f** Microscopy analysis (**e**) and quantification (**f**) of AlexaFluor 568-labelled trastuzumab (Tz-568) internalization in MDA-MB-361 cells silenced with SORLA (siSORLA #3) or scramble (siCTRL) siRNA at the indicated time points (mean ± s.e.m; n = 64, 77, 86 and 64 siCTRL cells and 111, 83, 103 and 87 siSORLA #3 cells at the 0, 15, 30 and 60 min time points, respectively, from two independent experiments; statistical analysis: Mann−Whitney test; a.u. arbitrary units). **g** Microscopy-based HER2 recycling assay in control or SORLA siRNA-treated MDA-MB-361 cells. Labelled HER2 recycling back to the plasma membrane was monitored over 30 min after an internalization step (45 min) and imaged with a confocal microscope. Ratio of surface/ internalized Tz-568 signal is displayed as box plots (n = 34 and 57 siCTRL cells and 45 and 47 siSORLA cells for 0 and 45 min time points, respectively, from two independent experiments; statistical analysis: Nonparametric Kruskal−Wallis). **h** Immunoblotting analysis of biotin-labelled cell-surface HER2 internalization in JIMT-1 cells overexpressing SORLA-GFP (or control GFP; GFP-CTRL), and quantification of internalized HER2 relative to total HER2 (data are mean ± s.d.; n = 4 independent experiments; statistical analysis: unpaired Student's t test). **i** Quantification of HER2 recycling rate (% return of internalized biotinylated cell-surface HER2 back to the plasma membrane after 10 min) in JIMT-1 cells transfected with GFP-CTRL or SORLA-GFP following 30 min of endocytosis (data are mean ± s.d.; n = 3 independent experiments; statistical analysis: unpaired Student's t test). Scale bars: 10 μm. Box plots represent median and IQR and whiskers extend to maximum and minimum values. Where micrographs are shown, these are representative of n = 3 independent experiments; ROI magnified region of interest

clinically relevant[39,40], we stained SORLA in a bladder cancer TMA of 199 patients. In this cancer type HER2 and SORLA levels correlated significantly (chi-square test, p = 0.0092), whereas there was no correlation between SORLA and EGFR levels (Fig. 6a, Supplementary Fig. 6a). This led us to test if SORLA also plays a role in the regulation of HER2 cell-surface levels, cell proliferation and cell sensitivity to CADs or in vivo tumour growth in this cancer model. Silencing of SORLA in 5637 cells, a bladder carcinoma cell line with a HER2-activating mutation[34], significantly inhibited their proliferation (Fig. 6b, c). In addition, cell-surface HER2 levels were consistently lower in SORLA-silenced 5637 cells than their control counterparts (Supplementary Fig. 6b), albeit these data did not reach statistical significance. Moreover, subcutaneous grafting of transiently SORLA-silenced 5637 cells (the strong anti-proliferative effect of shSORLA in vitro precluded sufficient propagation of stably silenced cells for in vivo experiments) in nude mice resulted in impaired tumour growth (mean tumour volume: control-silenced tumours, 78.7 mm³ and SORLA-silenced tumours, 47.7 mm³; p = 0.0461) (Fig. 6d, Supplementary Fig. 6c). Furthermore, SORLA-silenced 5637 tumours showed decreased proliferation, but not significantly increased apoptosis (Fig. 6e). Finally, in line with the breast cancer data, SORLA silencing sensitized 5637 cells to ebastine (Fig. 6f, g).

These results demonstrate that SORLA-mediated regulation of proliferation is not only restricted to HER2-amplified breast cancers, but is biologically important at least in HER2-driven urothelial cancer. It may be relevant to other neoplasms as well considering that data mining of the TCGA database revealed a significant positive correlation between *ErbB2* and *SORL1* expression in testicular germ cell tumours, cervical squamous cell carcinoma, endocervical adenocarcinoma, renal clear cell carcinoma, sarcoma and thymoma (Supplementary Fig. 6d).

## Discussion

Here we demonstrate that SORLA, a sorting protein previously not investigated in carcinomas, is highly expressed in many HER2-driven cancer cell lines, and that SORLA regulates HER2 subcellular localization by forming a complex with the receptor and coupling it to the recycling machinery (Fig. 7). We find that HER2 distribution between the plasma membrane and endosomes is highly heterogeneous in cancer cells; those with lower SORLA expression exhibit a trend of harbouring a substantial pool of intracellular HER2 at steady state. Furthermore, silencing of SORLA induces intracellular accumulation of HER2, and overexpressing SORLA triggers cell-surface localization of

HER2. Thus, HER2 undergoes rapid endosomal trafficking and recycling, the kinetics of which are regulated by SORLA-HER2 association. The fact that SORLA depletion (1) dramatically reduces proliferation of HER2-driven bladder and breast cancer cells in vitro and in vivo, (2) alters HER2 signalling, (3) interferes with lysosome integrity and (4) provides additive pro-apoptotic effects in combination with a clinically well-tolerated and widely used lysosome-accumulating drug provides new insight into the pathophysiology and targetability of HER2-driven oncogenesis.

Previous studies have shown discrepant results regarding HER2 trafficking. While some studies have shown that HER2 is resistant to internalization[13,16,17], others have suggested rapid recycling of HER2 back to the plasma membrane[22,23]. In both scenarios, HER2 is mainly restricted to the plasma membrane where it can associate with signalling platforms to drive the proliferation and tumorigenesis of cancer cells. Here we investigated the localization of HER2 in six different HER2-amplified breast cancer cell lines and found very different patterns of localization. Our observation of HER2 overlapping with EEA1- and VPS35-positive endosomes in some of the cell lines suggests that at least a pool of HER2 moves back to the plasma membrane via retromer-dependent vesicle trafficking[41]. The heterogeneity in the subcellular localization of HER2 is interesting, since it might reflect the functions of HER2 and the efficiency of therapeutic targeting of HER2. Indeed, in our study, anti-HER2 therapy (lapatinib, trastuzumab)-resistant JIMT-1, HCC1954 and MDA-MB-361 cell lines displayed more intracellular HER2 compared to the therapy-sensitive BT474, HCC1419 and SKBR3 cell lines with predominantly plasma membrane-localized HER2. This is also in line with two recent studies, published during the course of our investigation, that indicate a role for caveolin-1 in supporting HER2 internalization, and, in agreement with our findings, suggest that HER2 internalization and trafficking can be very different between different HER2-expressing cells[42,43]. Accordingly, the role of HER2 trafficking and the expression levels of specific trafficking proteins, such as caveolin-1 and SORLA, are important for the oncogenic activity of HER2 and most importantly for the response to anti-HER2 therapies. It is an intriguing possibility that the reduced viability of some HER2-dependent cancer cell lines upon depletion of trafficking proteins such as VPS35, Rab7 and LAMP1[44] might be related to alterations in HER2 trafficking.

The sorting functions of SORLA have been implicated in Alzheimer's disease and obesity[27–33]. We found that the ECD of SORLA associates either directly or indirectly with HER2, while the intracellular domain of SORLA is required for the correct endosomal and plasma membrane localization of this molecule in

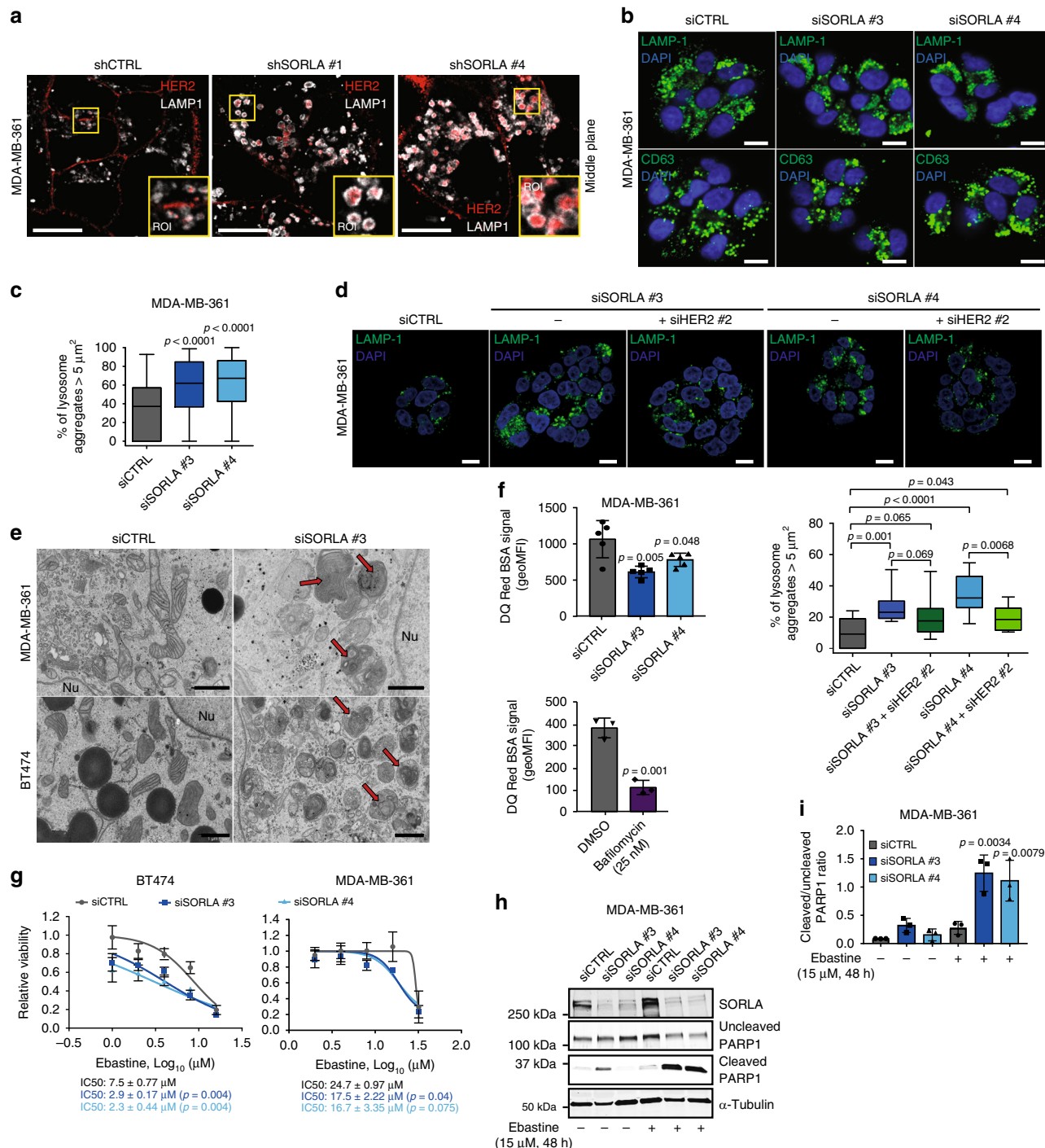

cancer cells. Importantly, re-expression of neither ECD-TM or TM-CD SORLA truncation mutants was sufficient to support proliferation of SORLA-silenced cells. This suggests that both the association of SORLA-ECD with HER2 and the correct sub-cellular localization of the SORLA-HER2 complex are necessary for proper SORLA function in supporting HER2-driven onco-genesis. Whether the observed coupling between SORLA-ECD and HER2 is mediated by direct binding or through other pro-teins remains to be investigated.

Lysosomal function is strongly linked to cellular fitness and it is especially important for the nutrient balance in rapidly growing cancer cells. One important finding of our work is that SORLA plays a major and unanticipated role in the maintenance of lysosome function in HER2-dependent cancer cells, but not in the

FGFR2-amplified, SORLA-positive MFM-223 cancer cells. The effects of SORLA depletion on lysosome function were two-fold. Lysosomes in SORLA-silenced cells showed perinuclear cluster-ing. In addition, we found that lysosomes were enlarged, dis-played an abnormal maturation defect-like appearance and had reduced proteolytic activity in SORLA-depleted cells. Strikingly, after SORLA silencing, HER2 accumulated into lysosomes with-out always being efficiently degraded (Fig. 7). The observed modest reduction in total HER2 protein levels was consistent, but varied in extent between experiments, which could be due to the parallel lysosomal defect. This is in stark contrast to the rapid lysosomal degradation of HER2 following treatment with gelda-namycin, which inhibits HSP90-CDC37-complex-mediated sta-bilization of HER2 on the plasma membrane[45,46] and suggests

**Fig. 4** Silencing SORLA induces HER2 accumulation in dysfunctional lysosomes. **a** Immunofluorescence imaging of LAMP1 (white) and HER2 (red) in shCTRL and shSORLA MDA-MB-361 cells ($n = 3$ independent experiments). **b** Immunofluorescence imaging of LAMP1 (green) and CD63 (LAMP3; green) in MDA-MB-361 cells (blue is DAPI) after scramble (siCTRL) or SORLA (siSORLA #3 and siSORLA #4) siRNA silencing ($n = 3$ independent experiments). **c** Quantification of late endosomes/lysosome aggregation after SORLA silencing in MDA-MB-361 cells. LAMP1-positive structures $\geq 5\,\mu m^2$ were considered as lysosome aggregates ($n = 73$ siCTRL, 79 siSORLA #3 and 67 siSORLA #4 cells from three independent experiments; statistical analysis: Mann−Whitney test). **d** Immunofluorescence imaging and quantification of lysosomal aggregation in MDA-MB-361 cells treated with the indicated siRNA ($n$ fields of view analysed, total cells = 11, 131 (siCTRL); 12, 182 (siSORLA #3); 11, 169 (siSORLA #3 + siHER2 #2); 10, 133 (siSORLA #4); 10, 121 (siSORLA#4 + siHER2 #2) from two independent experiments; statistical analysis: Mann−Whitney test). **e** Transmission electron microscopy imaging of lysosomes in siCTRL or siSORLA MDA-MB-361 and BT474 cells. Red arrows indicate the maturation defect in late endosome/lysosome structures ($n = 3$ technical replicates). **f** Flow cytometry analysis of the fluorescence signal in DQ Red BSA-loaded (24 h) MDA-MB-361 cells after scramble (siCTRL) or SORLA (siSORLA #3 and siSORLA #4) silencing. Cells loaded with DQ Red BSA (4 h) and treated with bafilomycin (or vehicle) are included as controls (bafilomycin blocks lysosome function) (mean ± s.d.; $n = 5$ independent experiments; statistical analysis: unpaired Student's $t$ test). **g** Cell viability assay to determine ebastine (48 h treatment) IC50 values in SORLA- or control-silenced BT474 and MDA-MB-361 cells (mean ± s.d.; $n = 12$, four technical replicates, three independent experiments; statistical analysis: unpaired Student's $t$ test). **h, i** Immunoblotting (**h**) and quantification (**i**) of cleaved PARP1 in ebastine-treated (15 µM, 48 h) siCTRL and siSORLA #3 or siSORLA #4 MDA-MB-361 cells. α-tubulin is a loading control (mean ± s.d.; $n = 3$ independent experiments; statistical analysis: unpaired Student's $t$ test). Scale bars: 10 µm (**a**, **b**, **d**) and 1 µm (**e**). Box plots represent median and IQR and whiskers extend to maximum and minimum values. Nu nucleus

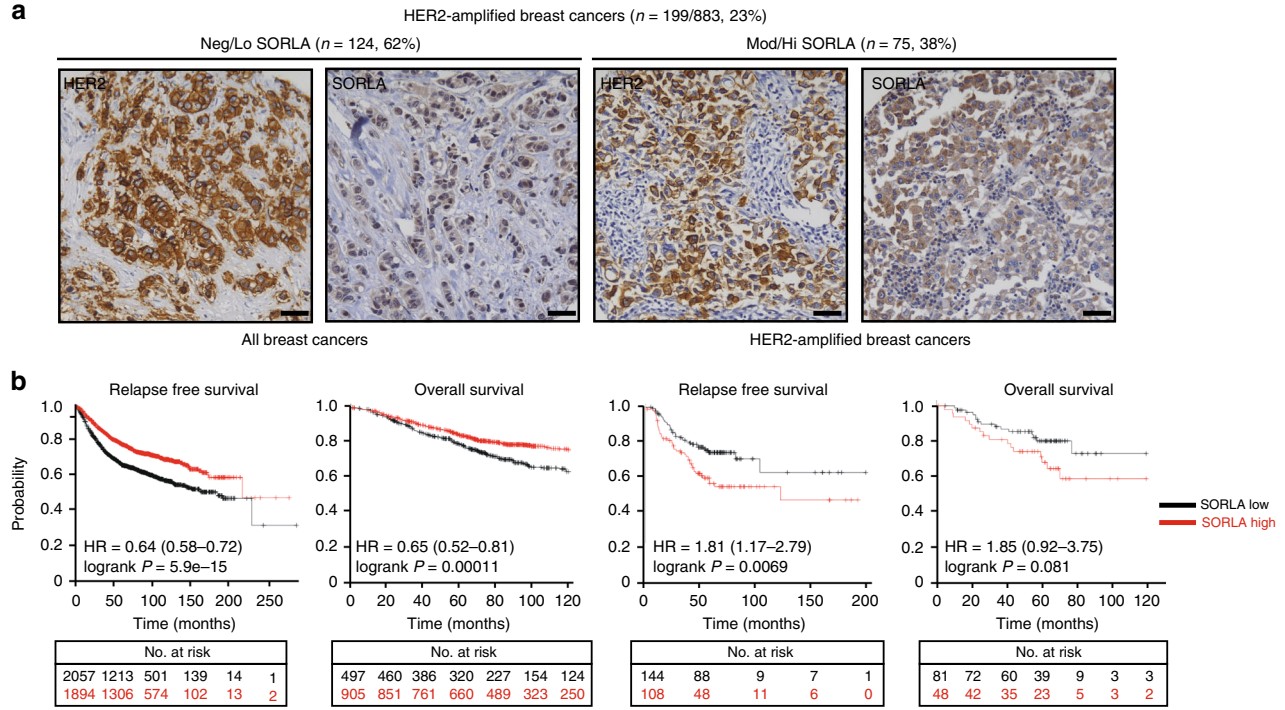

**Fig. 5** SORLA has a prognostic value in HER2-amplified breast cancer. **a** Immunohistochemical staining of SORLA and HER2 from a breast cancer tissue microarray (TMA; 883 patients in total). HER2-amplified tumours (199 patients) were categorized into negative/low (Neg/Lo) (staining intensity 0–1) and moderate/high (Mod/Hi) (staining intensity 2–3) groups. Numbers indicate staining intensity, 0 = negative, 1 = weak, 2 = moderate, 3 = high. **b** In silico biomarker assessment tool ((http://kmplot.com); including all datasets from 2010, 2012, 2014, 2017) analysis showing Kaplan−Meier plots of overall survival (OS; 10 years) and relapse-free survival (RFS; 20 years) of SORLA-high and SORLA-low patients (split by the best median cutoff) within all breast cancers (RFS $n = 3955$; OS $n = 1402$) and within HER2-amplified breast cancers (RFS $n = 252$; OS $n = 129$). Scale bars: 50 µm

that SORLA regulates HER2 in a fundamentally distinct way than HSP90. Possibly due to mistargeted localization and the lack of suitable signalling co-factors on late endosomes, HER2 signalling to the PI3K/AKT/mTOR proliferative pathway was decreased in SORLA-silenced breast cancer cells. Interestingly, double silencing of SORLA and HER2 led to a less enlarged and clustered lysosomal phenotype suggesting that lysosomes might be overwhelmed in the presence of excess intracellular HER2. Whether the SORLA silencing alone induces a mild lysosomal defect leading to HER2 accumulation in lysosomes and mislocalised HER2 further enhances lysosomal stress remains to be elucidated. The possible role of SORLA-dependent alterations in lipid and glucose metabolism[30–33] with respect to lysosomal dysfunction

also remains to be investigated. SORLA could additionally contribute to the oncogenic properties of cancer cells by regulating the trafficking of other cargo. For example, we find that cell-surface β1-integrin levels are reduced and integrin recycling is attenuated upon SORLA silencing.

We found that SORLA-silenced cells undergo apoptosis when exposed to low doses of the antihistamine ebastine, which belongs to a heterogeneous group of CADs with similar chemical features[47] and with a tendency to accumulate strongly in leaky lysosomes that are common in transformed cells[36,37]. CADs, including ebastine, have been utilized in several in vitro, pre-clinical and clinical trials to target the vulnerability of lysosomes in cancer[37]. Importantly, we found that both anti-HER2

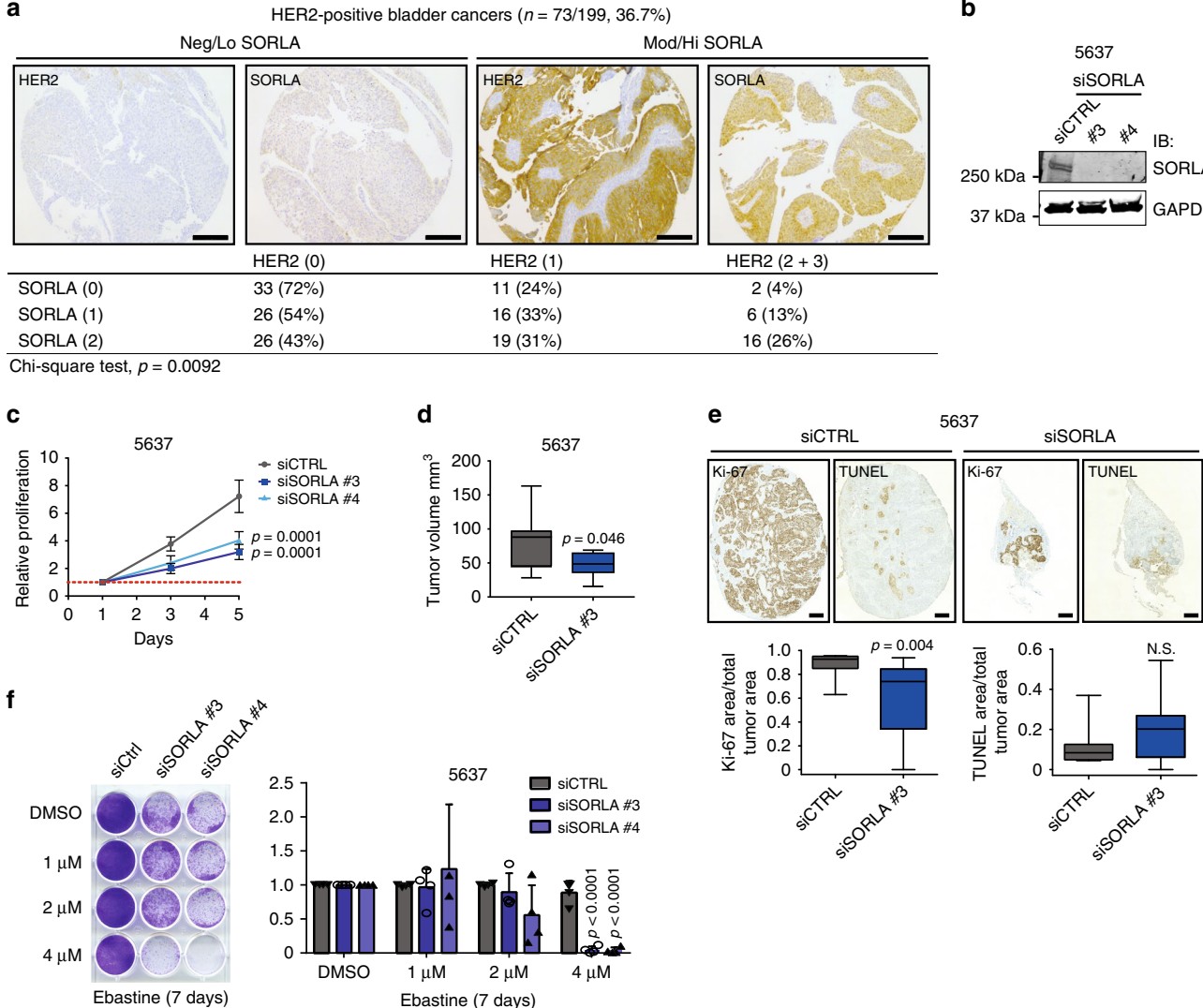

**Fig. 6** SORLA expression correlates with HER2 and promotes tumorigenesis in bladder carcinoma. **a** Immunohistochemical staining of a bladder cancer TMA and correlation analyses of SORLA and HER2 levels. Numbers indicate staining intensity, 0 = negative, 1 = weak, 2 = moderate, 3 = high. **b** Western blot analysis of SORLA levels in siCTRL and siSORLA (siSORLA#3, siSORLA#4) 5637 bladder cancer cells. GAPDH is a loading control. **c** Analysis of siCTRL and siSORLA (siSORLA#3, siSORLA#4) 5637 cell proliferation (mean ± s.d; $n = 4$ independent experiments; statistical analysis: two-way ANOVA). **d** Analysis of tumour growth of subcutaneously injected 5637 cells, with transient SORLA (siSORLA #3) or scramble (siCTRL) silencing, at day 29 in nude mice ($n = 9$ siCTRL and 10 siSORLA mice; statistical analysis: unpaired Student's t test). **e** Ki-67 and TUNEL staining of tumour samples prepared as described in **d** and quantifications displayed as box plots ($n = 9$ siCTRL and 10 siSORLA mice; statistical analysis: Mann−Whitney test). **f** Colony formation assay with SORLA-silenced and control-silenced 5637 cells treated with different concentrations of ebastine for 7 days. Quantification of confluency was performed using ImageJ Colony Area Plug-in[49]. Results are shown as mean ± s.d. ($n = 4$ independent experiments; statistical analysis: unpaired Students t test). Scale bars: 200 μm (**a**), 500 μm (**e**). Box plots represent median and IQR and whiskers extend to maximum and minimum values

therapy-resistant and sensitive HER2-amplified breast cancer cells are susceptible to the combination of SORLA silencing and low doses of ebastine. Silencing SORLA alone mainly reduced proliferation, but when combined with ebastine triggered enhanced apoptosis. This additive effect could potentially be exploited by combining current anti-HER2 therapies with CADs, since breast cancer cells and patient-derived xenografts resistant to an anti-HER2 drug-antibody conjugate were recently shown to possess dysfunctional lysosomes[48].

In conclusion, we have discovered that SORLA regulates the subcellular localization of HER2 and that dynamic recycling is essential for the oncogenic fitness of HER2. The endosomal trafficking of HER2 could provide new rationale for designing targeted therapies and understanding the resistance mechanisms induced by the current HER2-targeting therapies.

## Methods

**Subcutaneous and ductal carcinoma in situ (DCIS) xenograft models**. For subcutaneous (s.c.) tumours, $2 \times 10^6$ siCTRL- and siSORLA #3-treated 5637 bladder cancer cells were injected s.c. (100 μl volume; 50% Matrigel, 50% PBS) at the flank of 6-week-old nude mice. Mice were sacrificed 29 days later, and tumours were dissected, fixed in 10% formalin, and processed for paraffin sections with standard protocols. Sections were stained with haematoxylin-eosin (HE) and with immunohistochemistry (IHC) antibodies against proliferation (Ki67) and apoptosis (TUNEL) markers. Sections were imaged with a pannoramic slide scanner.

For intraductal tumour xenografts, MDA-MB-361 shCTRL and shSORLA-silenced breast cancer cells were resuspended in PBS ($2.5 \times 10^4$ cells/μl). Trypan blue (0.1%) was added to the cell solution to visualize successful injection. Eight- to ten-week-old female NOD.SCID mice were medicated with Temgesic for analgesia and anesthetized with isoflurane. After removal of abdominal hair, the tip of the abdominal (fourth) mammary glands was carefully snipped, and 4 μl ($1 \times 10^5$ cells) of cell suspension was injected into the mammary ducts. A 30 G Hamilton syringe with 50-μl capacity and a blunt-ended needle was used for the injection under a stereomicroscope. Post-operatively, the mice were further dosed with Rimadyl for

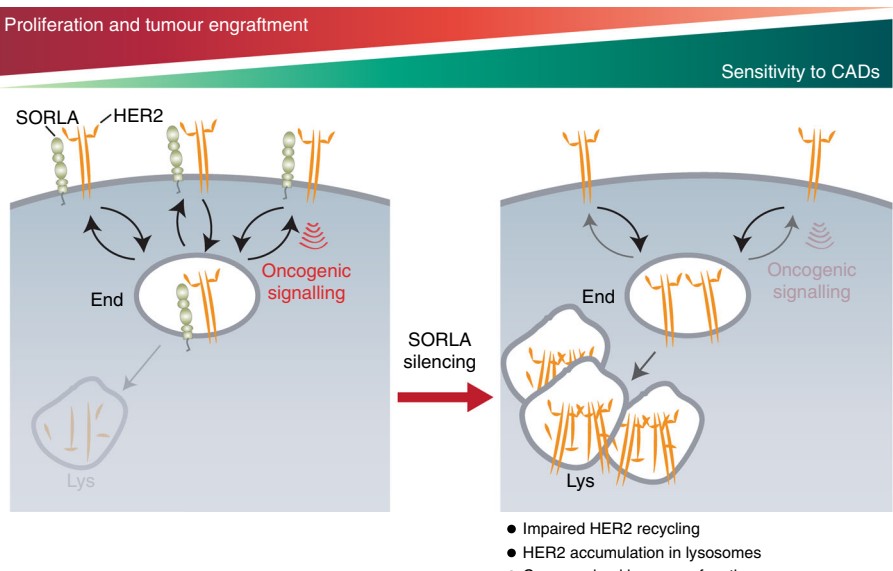

**Fig. 7** Schematic illustrating the role of SORLA in the oncogenic fitness of HER2 in cancer cells. SORLA, through interactions at its extracellular domain, is in a complex with HER2 and co-traffics with HER2, facilitating HER2 recycling to the plasma membrane to support HER2 downstream signalling. In the absence of SORLA, HER2 becomes localized to enlarged, partially dysfunctional lysosomes resulting in defective HER2 signalling and increased sensitivity to cationic amphiphilic drugs (CADs) like ebastine. End endosome, Lys lysosome

pain relief. Mice were sacrificed 10 weeks after tumour inoculation, and abdominal mammary glands were dissected. Mammary gland whole mounts were prepared on object glasses and fixed in Carnoy's medium (60% EtOH, 30% chloroform, 10% glacial acetic acid) overnight (o/n) at +4 °C. After rehydration in decreasing EtOH series and staining with carmine alum (0.2% carmine, 0.5% aluminium potassium sulphate dodecahydrate) o/n at room temperature (RT), samples were dehydrated and cleared in xylene for 2–3 days. Samples were mounted in DPX Mountant for histology (Sigma) and images were taken with Zeiss SteREO Lumar V12 stereomicroscope (NeoLumar ×0.8 objective, Zeiss AxioCam ICc3 colour camera). All images per gland were combined automatically into a mosaic picture with PhotoShop. DCIS area per gland was quantified in ImageJ.

All animal studies were ethically performed and authorized by the National Animal Experiment Board and in accordance with The Finnish Act on Animal Experimentation (Animal license number ESAVI-9339-04.10.07-2016).

**Cell lines and cell culture**. MDA-MB-361 cells (ATCC, HTB-27) were grown in Dulbecco's modified essential medium (DMEM; Sigma-Aldrich, D5769) supplemented with 20% foetal bovine serum (FBS; Sigma-Aldrich, F7524), 1% vol/vol penicillin/streptomycin (Sigma-Aldrich, P0781-100ML) and L-glutamine. BT474 (ATCC, HTB-20) and 5637 (ATCC, HTB-9) cells were grown in RPMI-1640 (Sigma-Aldrich, R5886) supplemented with 10% FBS, 1% vol/vol penicillin/streptomycin and L-glutamine. JIMT-1 (DSMZ, ACC 589), HCC1954 (ATCC, CRL-2338), HCC1419 (ATCC, CRL-2326), MCF7 (ATCC, HTB-22), MDA-MB-231 (ATCC, HTB-26), MDA-MB-436 (ATCC, HTB-130) and MFM-223 (DSMZ, ACC-422) were grown in DMEM supplemented with 10% FBS, 1% penicillin/streptomycin and L-glutamine. MCF10A (ATCC, CRL-10317) and MCF10A DCIS.com (provided by Prof. J.F. Marshall, Barts Cancer Institute, Queen Mary University of London, London, UK) were grown in DMEM/F12 (Invitrogen, #11330-032) supplemented with 5% horse serum (Invitrogen#16050-122), 20 ng/ml human epidermal growth factor (Sigma-Aldrich, E9644), 0.5 mg/ml hydrocortisone (H0888–1G; Sigma-Aldrich), 100 ng/ml insulin (Sigma-Aldrich, I9278-5ML) and 1% vol/vol penicillin/streptomycin. All cells were regularly tested for mycoplasma infection and were grown at +37 °C, 5% $CO_2$ until 70–80% confluence before being detached and replated. The medium was changed every 3 days. T24 (ATCC, HTB-4) and SKBR3 (ATCC, HTB-30) cells were grown in McCoy's 5a Medium Modified (Sigma-Aldrich, M8403-500ML) supplemented with 10% FBS, 1% penicillin/streptomycin and L-glutamine, and the patient-derived bladder carcinoma line (provided by Dr. P. Taimen, Turku University Hospital, Turku, Finland) was grown in F-medium (3:1 (v/v) DMEM: F-12 Nutrient Mixture (Ham): DMEM (Invitrogen) supplemented with 5% FCS, 0.25 µg/ml hydrocortisone, 5 mg/ml insulin, 10 mg/ml gentamicin, 250 µg/ml fungizone, 8.6 ng/ml cholera toxin, 125 ng/ml EGF) collected from irradiated Swiss 3T3 J2 mouse fibroblast feeder cell cultures and supplemented with 10 µM ROCK inhibitor (Y-27632, Enzo Life Sciences, Lausen, Switzerland).

**Antibodies**. The antibodies used are described in Supplementary Table 1.

**Generation of lentiviral shRNA and SORLA-GFP particles**. Lentiviral particles containing sequences encoding shRNA against SORLA and GFP or a control scramble shRNA sequence and GFP or particles encoding for SORLA-GFP or GFP alone were generated in the 293FT packaging cell line (complete medium: high glucose DMEM, 10% FBS, 0.1 mM NEAA, 1 mM MEM Sodium Pyruvate, 6 mM L-glutamine, 1% penicillin/streptomycin and 0.5 mg/ml Geneticin) by transient transfection of transfer vector (#TL309181A SORL1 SR031650, #TL309181B SORL1 SR031650, #TL309181C SORL1 SR031650, #TL309181D SORL1 SR031650, or scramble control (#TR30021, Origene), second-generation packaging plasmid-psPAX2 (Addgene #12259) and envelope vector-pMD2 (Addgene #12260) with the ratio (7:2:1) using calcium-phosphate precipitation method[50]. Seventy-two hours post transfection, medium containing viral vectors was collected, concentrated for 2 h by ultracentrifugation (26,000 × g) in a swing-out rotor SW-32Ti (Beckman Coulter, Brea, CA, USA), resuspended in residual medium and flash frozen in liquid nitrogen. Functional titre was evaluated in 293FT cells by FACS (BD LSRFortessa, Becton Dickinson).

**Lentiviral transduction to generate stable cell lines**. To generate stable silenced cell lines $4 \times 10^5$ BT474 or MDA-MB-361 cells were seeded on 10 cm dishes and transduced 24 h later with MOI 47 of lentivirus in a low volume of full media. To obtain stable overexpression of SORLA-GFP or GFP, $8 \times 10^4$ JIMT-1 or MDA-MB-361 cells were seeded in a 24-well plate and transduced 24 h later with MOI 60. Medium containing viral particles was removed 16 h later. Cells expressing GFP, indicative of lentiviral integration were collected by fluorescence-assisted cell sorting (BD FACSaria II cell sorter, Becton Dickinson, Franklin Lakes, NJ, USA) with a gating strategy to obtain medium expression.

**Transient transfections**. For transient protein expression, Lipofectamine 3000 (Invitrogen, P/N 100022052) and P3000 enhancer reagent (Invitrogen, P/N100022058) were used according to the manufacturer's instructions. Cells were transfected with plasmids 24 h prior to experiments. Transient siRNA transfections were performed using Lipofectamine RNAiMAX reagent (Invitrogen, P/N 56532) according to the manufacturer's instructions. SORLA-targeting siRNAs were ON-TARGETplus obtained from Dharmacon—siSORLA #1 (J-004722-08), siSORLA #2 (J-004722-06), siSORLA #3 (J-004722-07), siSORLA #4 (J-004722-05). For rescue experiments, siRNA against the 3′UTR end of SORLA was obtained from Qiagen (siSORLA 3′UTR, SI05039888). HER2-targeting siRNAs were ON-TARGETplus obtained from Dharmacon (siHER2 #2, J-003126-17; siHER2 #4, J-003126-20). For controls, Allstars negative control (Qiagen, Cat. No. 1027281) was used. SiRNA concentrations used ranged between 20 and 40 nM and cells were transfected with siRNAs 48 h prior to experiments.

The corresponding oligonucleotide sequences can be found in Supplementary Table 2.

**Western blot analysis**. Protein extracts were separated using SDS-PAGE under denaturing conditions (4–20% Mini-PROTEAN TGX Gels) and were transferred to nitrocellulose membranes (Bio-Rad Laboratories). Membranes were blocked with

5% milk-TBST (Tris-buffered saline and 0.1% Tween 20) and incubated with the indicated primary antibodies overnight at +4 °C. Primary antibodies were diluted in blocking buffer (Thermo, StartingBlock (PBS) blocking, #37538) and PBS (1:1 ratio) mix and incubated overnight at +4 °C. Primary antibody dilutions used ranged from 1:500 to 1:1000. After primary antibody incubation, membranes were washed three times with TBST and incubated with fluorophore-conjugated secondary antibodies diluted (1:1000) in blocking buffer at RT for 1 h. Membranes were scanned using an infrared imaging system (Odyssey; LI-COR Biosciences). The following secondary antibodies were used: donkey anti-mouse IRDye 800CW (LI-COR, 926-32212), donkey anti-mouse IRDye 680RD (LI-COR, 926-68072), donkey anti-rabbit IRDye 800CW (LI-COR, 926-32213) and donkey anti-rabbit IRDye 680RD (LI-COR, 926-68073). All original western blots for the manuscript can be found in Supplementary Fig. 7.

**Co-immunoprecipitations.** Cells were lysed in IP-lysis buffer (0.5% Triton X-100, 10 mM Pipes, pH 6.8, 150 mM NaCl, 150 mM sucrose, 3 mM MgCl$_2$ and complete protease and phosphatase inhibitors (Mediq; Roche)), cleared by centrifugation (13,226 × g, 10 min, 4 °C), and incubated with GFP-trap beads to pull-down GFP-proteins (Chromotek; gtak-20) for 1 h at +4 °C or with mouse anti-HER2 antibody (1 μg/sample; ThermoScientific, MA5-14057) or isotype matching IgG control antibody at +4 °C overnight. Antibody complexes were bound to 0.5% BSA pre-blocked protein-G sepharose beads for 1 h at 4 °C. Complexes bound to the beads were isolated using 1000 × g 3 min centrifugation, washed three times with washing buffer (20 mM Tris-HCl (pH 7.5), 150 mM NaCl, 1 % NP-40; 500 μl) and eluted in sample buffer. Input and precipitate samples were analysed by western blotting. Primary antibodies were incubated overnight at +4 °C. Mouse anti-HER2 (ThermoScientific; MA5-14057), rabbit anti-GFP (Molecular Probes; A11122) and mouse anti-SORLA (anti-LR11, BD Transduction Lab; 612633) diluted 1:1000 in 5% milk in TBST were used followed by the appropriate IRDye conjugated secondary antibodies.

**Immunofluorescence staining and imaging.** Cells were plated on μ-Slide 8-well (Ibidi, 80826) or in some cases in μ-dish 3.5 mm dishes (Ibidi, 80136). Cells were fixed with 4% paraformaldehyde (PFA) 10 min at RT, quenched with 50 mM NH$_4$Cl for 15 min at RT, blocked and permeabilized with 30% horse serum in PBS + 0.3% Triton X-100 for 10 min at RT and incubated with primary antibodies diluted in 30% horse serum overnight at +4 °C. Staining was performed using antibodies against HER2 (trastuzumab, Roche, 0.15 μg/ml; or mouse monoclonal antibody, ThermoScientific, MA5-14057, 1:300 dilution), LAMP1 (Santa Cruz; SC-20011 (H4A3), dilution 1:50), SORLA (rabbit monoclonal, CM Petersen Lab, Århus University, dilution 1:300), EEA-1 (goat polyclonal, Santa Cruz; sc-6415, dilution 1:50), VPS35 (goat polyclonal, Abcam; ab10099, dilution 1:300), CD63 (mouse mAb, Hybridoma Bank; H5C6, dilution 1:300) and Rab11 (rabbit polyclonal, Cell Signaling Technology, #5589, dilution 1:100). After several washes, appropriate secondary antibodies (donkey anti-mouse AlexaFluor 488 (Life Technologies, A21202), donkey anti-rabbit AlexaFluor 488 (Invitrogen), goat anti-human AlexaFluor 568 (Invitrogen, A21090), donkey anti-goat AlexaFluor 647 (Invitrogen, A21447), donkey anti-mouse AlexaFluor (Invitrogen, A31571)) diluted 1:300 in 30% horse serum were added together with DAPI (1:1000) for 1 h at RT. After PBS washes, samples were imaged right away or stored at +4 °C in the dark. Imaging was performed either with a Carl Zeiss LSM780 laser scanning confocal microscope or a 3i CSU-W1 spinning disk confocal microscope with Hamamatsu CMOS (×63 objective).

**Transmission electron microscopy.** Cells were fixed in 5% glutaraldehyde in 0.16 M s-collidine buffer, pH 7.4. The samples were post-fixed for 2 h with 1% OsO4 containing 1.5% potassium ferrocyanide, dehydrated with a series of increasing ethanol concentrations and embedded in 45359 Fluka Epoxy Embedding Medium kit. 70-nm sections were cut with an ultramicrotome and stained with 1% uranyl acetate and 0.3% lead citrate. The sections were examined with a JEOL JEM-1400 Plus transmission electron microscope.

**Live-cell imaging.** Lentiviral transduced SORLA-GFP-expressing MDA-MB-361 cells were kept on ice and washed twice with ice-cold PBS. Alexa-568-conjugated trastuzumab (0.15 μg/ml) in Hank's Balanced Salt Solution was incubated with the cells on ice for 1 h protected from light. The cells were washed twice with ice-cold PBS before addition of pre-warmed culture media supplemented with 5% HEPES (without serum). Imaging was performed with Deltavision OMX V4 total internal reflection microscopy (TIRFM; GE Healthcare) every 250 ms with a ×60/1.49 objective (Olympus TIRF objective).

**Proliferation assay.** Cells, transfected as indicated, were plated on 96-well plates (3000 cells/well) in a volume of 100 μl. After 1, 4 and 8 days of cell growth, 10 μl/well of WST-8 reagent (cell counting kit 8, Sigma-Aldrich, 96992) was added and absorbance at 450 nm was measured by a plate reader (Thermo, Multiscan Ascent) after 1–2 h of incubation at +37 °C with 5% CO$_2$. Medium without cells was used as background and the A450 of background was subtracted from the samples. Relative proliferation was calculated by normalizing the A450 values of 4 and 8 days to 1 day A450 values. For drug sensitivity assays cells were treated the

following day after plating with increasing concentrations of ebastine (BT474: 2, 4, 8, 16, 32 μM and MDA-MB-361: 2, 4, 8, 16, 32, 64 μM). DMSO was used as a control and proliferation was measured after 48 h.

**Analysis of SORLA and HER2 cell-surface levels.** Cells were detached by HyQtase, fixed with 4% PFA (in PBS for 15 min at RT), and washed before labelling with 1:1000 dilution of a mouse anti-HER2 antibody (9G6, Abcam; Ab16899), or 1:100 dilution of a goat anti-SORLA primary antibody[28]. After washing with PBS, samples were incubated with donkey anti-mouse AlexaFluor 647, donkey anti-mouse AlexaFluor 488 or donkey anti-goat AlexaFluor 488 secondary antibody (dilution 1:300 in PBS; all from Invitrogen), washed with PBS and analysed with LSRFortessa (BD Biosciences). Data analysis was performed with Flowing software version 2 (Cell Imaging Core of the Turku Bioscience Centre). The geometric mean of the fluorescence intensity from cells labelled with secondary antibody alone was used as background and subtracted from the stained samples. For normalization, the background corrected values were divided by the sum of all signals within one independent experiment.

Representative raw flow cytometry data can be found in Supplementary Fig. 8.

**DQ Red BSA assay.** For microscopy assays, silenced cells were plated on ibidi 8-well μ-slide 72 h post transfection. On the next day, the medium was replaced with DQ Red BSA (ThermoScientific, D12051)-containing medium (25 μg/ml, 200 μl/well) and cells were incubated for 48 h at 37 °C with 5% CO$_2$, the cells were washed, fixed with 4% PFA for 10 min at RT, washed again and imaged with a Carl Zeiss LSM780 laser scanning confocal microscope.

For flow cytometry assays, the cells were silenced and loaded with DQ Red BSA (25 μg/ml, 1 ml per well in six-well plates for 24 h) as above. When indicated, 25 nm bafilomycin (Calbiochem, 196000-10UG) or DMSO only was added to DQ Red BSA and the loading was shortened to 4 h. The loaded cells were detached with HyQtase, washed, fixed with 4% PFA for 15 min at RT, washed, and analysed using LSRFortessa (BD Biosciences) and Flowing software. Fluorescence signals from unstained cells (background) were subtracted from the DQ Red BSA signals and geometric means were plotted.

Representative raw flow cytometry data can be found in Supplementary Fig. 8.

**RNA extraction, cDNA synthesis and qPCR.** Cells were lysed in RA lysis buffer and RNA was extracted according to the manufacturer's instructions (NucleoSpin RNA extraction kit, Macherey−Nagel, 740,955.5). RNA concentration was measured by NanoDrop Lite (Thermo). Complementary DNA (cDNA) was synthesized using a high-capacity cDNA Reverse Transcription Kit (Applied Biosystems) according to the manufacturer's instructions. Quantitative real-time PCR reactions with TaqMan probes were performed according to the manufacturer's instructions (Thermo/Applied Biosystems, TaqMan™ Universal Master Mix II, 4440040). The following TaqMan probes (ThermoScientific, 4331182) were used: ErbB2 (Hs01001580_m1). Relative quantification of gene expression values were calculated using the ddCt method[51].

**Biotin-based HER2 endocytosis assay.** HER2 endocytosis was measured using a cell-surface biotinylation-based assay as previously described[52]. Briefly, JIMT-1 cells expressing GFP-CTRL or SORLA-GFP were grown to 80% confluence, placed on ice, and washed once with cold PBS. Cell-surface proteins were labelled with 0.5 mg/ml of EZ-link cleavable sulfo-NHS-SS-biotin (#21331; Thermo Scientific) in Hanks' balanced salt solution (H9269; Sigma) for 30 min at 4 °C. Unbound biotin was removed by washing with cold Hanks' balanced salt solution. Thereafter, the cells were allowed to internalize receptors in pre-warmed 10% serum-containing medium at +37 °C for the indicated times. Internalization was stopped by transferring the cells to ice and adding cold medium. The remaining biotin on the cell surface was removed with 60 mM MesNa (63705; sodium 2-mercaptoethane-sulfonate: Fluka) in MesNa buffer (50 mM Tris-HCl [pH 8.6], 100 mM NaCl) for 30 min at 4 °C, followed by quenching with 100 mM iodoacetamide (IAA, Sigma) for 15 min on ice. To detect the total surface biotinylation, plates were left on ice after the biotin labelling and MesNa treatment was omitted. Cells were then washed with PBS, scraped in lysis buffer (50 mM Tris pH 7.5, 1.5 % Triton X-100, 100 mM NaCl, complete protease and phosphatase inhibitors (Mediq; Roche)) at 4 °C for 20 min. After clarification by centrifugation (14,000 × g, 10 min, 4 °C), HER2 was immunoprecipitated from the supernatants with appropriate antibodies and protein G sepharose beads (17-0618-01; GE Healthcare). The immunoprecipitates were eluted in non-reducing Laemmli sample buffer and subjected to western blotting as described above. Biotinylated (internalized) HER2 and total receptor levels were detected by immunoblotting with horseradish peroxidase (HRP)-conjugated anti-biotin antibody (#7075; Cell Signaling Technology) and receptor-specific antibodies, respectively. Enhanced chemiluminescence-detected biotin and receptor signals were quantified as integrated densities of protein bands with ImageJ (v. 1.43 u), and each biotin signal was normalized to the corresponding receptor and total biotin signals.

**HER2 recycling assay in JIMT-1 SORLA-GFP cells.** HER2 recycling was measured using a cell-surface biotinylation-based assay as previously described[52]. Cell-surface proteins were biotinylated and allowed to be internalized as described

above for 30 min. After the first MesNa/IAA treatment, the internalized biotinylated fraction of receptors was then chased for the indicated time points by returning cells to 37 °C with pre-warmed medium containing 10% FBS, followed by a second MesNa/IAA treatment to cleave biotin from the recycled cell-surface proteins. HER2 was immunoprecipitated and analysed for biotinylation and total levels as in the endocytosis assay.

Alternatively, biotinylated HER2 was determined by capture ELISA as described previously[53] using trastuzumab (Herceptin) or c-erbB-2 (A0485, Dako). Briefly, 50 µl of the antibody (5 µg/ml diluted in 0.05 M $Na_2CO_3$ pH 9.6) was added into 96-well plates (Maxisorb, Thermo Scientific Nunc-Immunoplate) and incubated o/n at +4 °C. Plates were washed with 0.1% PBS-T and blocked with 5% BSA (41-00-410, First Link) in 0.1% PBS-T for 2 h at RT. The plate was washed extensively with 0.1% PBS-T and the biotinylated lysate was added and incubated o/n at +4 °C. On the following day the plate was washed extensively with 0.1% PBS-T and the secondary antibody (Pierce High Sensitivity Streptavidin-HRP 21130) was added in 1:1000 in 1% BSA in 0.1% PBS-T and incubated for 2 h at RT. The plate was then washed extensively with 0.1% PBS-T and once with PBS and 50 µl of the detection reagent (0.8 mg/ml of O-Phenylenediaminedihydrochloride (Sigma) in detection buffer (25.4 mM $Na_2HPO_4$, 12.3 mM citric acid, pH 5.4) supplemented with 0.2 µl/ml of $H_2O_2$ was added. Incubation was continued until colour formation (2–15 min) and stopped by addition of 50 µl of 8 M $H_2SO_4$ and absorbance was read at 490 nm.

**Imaging-based HER2 internalization and recycling assay**. MDA-MB-361 cells were plated on ibidi 35 mm µ-dishes after 72 h of silencing. The following day, cells were washed once with PBS and incubated with Alexa-568-conjugated trastuzumab (Tz-568; 0.15 µg/ml) in cold Hank's Balanced Salt Solution on ice for 1 h protected from light. Internalization was triggered with a temperature shift by adding pre-warmed serum-free media to the cells and incubating at +37 °C for the indicated times. The 0 min sample serves as the no internalization control. The antibody internalization was stopped by washing with cold PBS and fixing with 4% PFA for 10 min at RT. Fixed cells were washed and stored at +4 °C in the dark until imaging with 3i spinning disk confocal microscope. The recycling experiments were performed in DMEM + 10% FBS, and the antibody internalization (Tz-568; 2 µg/ml) was allowed to proceed for 45 min at +37 °C. This was followed by an acid wash (0.2 M acetic acid, 0.5 M NaCl, pH 2.5) to remove surface-bound antibodies. After three washes with PBS, the cells were overlaid with DMEM + 10% FBS and incubated for 0 or 30 min at +37 °C to allow antibody recycling and followed by fixation with 4% PFA. Imaging was performed with Leica SP5 confocal microscope.

**Immunohistochemistry, Ki-67 and TUNEL labelling**. Formalin-fixed, paraffin-embedded tissue samples were cut to 4 µm sections, and deparaffinized and rehydrated with standard procedures. For immunohistochemistry of mouse xenografts, heat-mediated antigen retrieval was done for all samples in citrate buffer (pH 6). For Ki-67 labelling, sections were washed with washing buffer (Tris-HCl 0.05 M pH 7.6, 0.05 % Tween 20) and Normal Antibody Diluent (NABD; Immunologic, BD09-125) and incubated with a Ki-67 antibody (14-5698, eBioscience, clone SolA15, diluted 1:2000) for 1 h. After washes, samples were incubated in 3% $H_2O_2$ Tris-HCl for 10 min and washed again. Samples were incubated for 30 min with Rat probe (Biocare Medical rat on mouse HRP-polymer RT517 –kit), washed, and further incubated with R-O-M HRP-polymer (Biocare Medical rat on mouse HRP-polymer RT517 –kit) for 30 min. After washes, DAB solution (DAKO K3468) was added for 10 s followed by washing. After counter-stain with Mayer's HTX, slides were dehydrated, cleared in xylene and mounted.

For TUNEL staining of apoptotic cells, endogenous peroxidase activity was blocked with incubation in 3% $H_2O_2$ in PBS for 15 min. The reaction mix containing recombinant terminal transferase (Cat. No. 03 333 566 001, Roche), $CoCl_2$ and Biotin-16-dUTP (11 093 070 910, Roche) in TdT buffer (1 M potassium cacodylate, 3% BSA in PBS, pH 6.6) was applied on slides and incubated for 1 h at +37 °C in a humidified chamber. As a positive control, one section was incubated for 30 min in DNAse solution at +37 °C before TUNEL staining, and as a negative control, the reaction mix was added without the transferase. Then the samples were incubated with End reaction solution (300 mM NaCl, 30 mM sodium citrate) for 30 min at RT and washed. Blocking was done with 3% BSA in PBS 45 min at RT after which the samples were incubated with ExtrAvidine (1:500 dilution in 1% BSA/PBS) for 30 min at +37 °C in a humidified chamber. After PBS washes, DAB solution (DAKO K3468) was added for 10 s followed by washing. After counterstain with Mayer's HTX, the slides were dehydrated, cleared in xylene, mounted and imaged with a Pannoramic 250 Slide Scanner (3DHISTECH Ltd).

For bladder cancer TMA immunohistochemistry, BenchMark XT automated IHC/ISH slide staining system (Ventana Medical Systems, Inc.) with Cell Conditioning Solution (CC1) as a pretreatment was used for anti-HER2/neu (rabbit monoclonal, Ventana clone 4B5, 24 min incubation) and anti-EGFR (rabbit monoclonal, Ventana clone 5B7, 32 min), followed by UltraView Universal DAB Detection Kit (Ventana). For SORLA staining, antigen retrieval pre-treatment was first done by microwaving the slides in citrate buffer (pH 6). Lab Vision autostainer (Thermo Fisher Scientific) with SORLA antibody (rabbit polyclonal, Atlas Antibodies HPA 031321; 1:300 dilution) was used and primary antibodies were detected with PowerVision Poly-HRP anti-mouse/anti-rabbit IHC system (Leica BioSystems). Finally, the slides were counterstained with haematoxylin.

IHC staining intensities of each tissue core were visually scored for SORLA as 0 (negative), 1 (weak to moderate), or 2 (strong), and for HER2 and EGFR as 0 (negative), 1 (weak), 2 (moderate) or 3 (strong). SORLA and HER2/EGFR staining was evaluated independently by two readers and the mean and maximum values of three tissue cores were determined for each patient. In the final statistical analysis, maximum scores for all the staining were used.

For the FinHer Breast Cancer tissue microarray immunohistochemistry, deparaffinized tissues were incubated in hydrogen peroxidase to block endogenous peroxidase activity, and antigen was unmasked by using sodium citrate (10 mmol/L, pH 6.0) and 2100 Antigen Retriever instrument (Aptum Biologics Ltd., Southampton, UK). SORLA antibody (rabbit polyclonal, Atlas Antibodies HPA 031321; 1:200 dilution) was diluted in a Normal antibody Diluent (Immunologic, Duiven, The Netherlands), and incubated on slides for 1 h at RT. Binding of the primary antibody was detected and visualized by using the BrightVision Poly-HRP anti-Rabbit kit (Immunologic) and 3,3′-diaminobenzidine (ImmPACT™ DAB, Vector Laboratories, Burlingame, CA, USA) following the manufacturer's recommendations. The slides were counterstained with Mayer's haematoxylin.

**Tissue microarray construction**. Bladder cancer TMA construction was approved by the Research Ethics Board of the Hospital District of Southwest Finland (1.8.2006/301). FFPE tissue samples from consecutive 199 patients who underwent radical cystectomy in Turku University Hospital between 1985 and 2005 were used and three 1 mm tissue cores per patient were punched for TMA. Average age at cystectomy was 64 and none received neoadjuvant therapies. FinHer series TMA has been described previously[1].

**Quantification of intracellular HER2 levels**. Several fields were randomly imaged with identical microscope settings. ImageJ was used for analysis and quantifications. Intracellular Tz-568 fluorescence signal was analysed from maximal intensity projections of six planes taken from the middle of the cell (determined by DAPI signal). Intracellular endogenous HER2 fluorescence intensity was measured from a single middle plane of the cell. Intracellular signal was quantified by manually gating the intracellular part of the cell and the intracellular signal was measured and normalized to the cell area. Results are pooled from two independent biological replicates.

**Quantification of late-endosome/lysosome aggregation**. SORLA-silenced and control-silenced cells were fixed and stained for LAMP1 (Santa-Cruz, SC-20011) and DAPI as described above. Cells were imaged with identical microscope settings with a Carl Zeiss LSM780 laser scanning confocal microscope. Image processing and quantifications were performed with the ImageJ software. The level of LAMP1-positive late-endosome/lysosome aggregation was quantified from the middle plane. To distinguish between individual late-endosome/lysosome aggregates, watershed segmentation was used before quantification. The areas of late-endosome/lysosome aggregates were quantified from a single cell. A LAMP1-positive area larger than 5 µm² was considered as aggregate. Total area of LAMP1-positive late endosomes/lysosomes was calculated based on LAMP1 staining as well as the area of aggregates larger than 5 µm². The percentage of lysosomal aggregation was determined by the ratio of area of aggregates larger than 5 µm² to total LAMP1-positive late-endosome/lysosome area.

**Analysis of Ki-67 and TUNEL staining in 5637 cell xenografts**. Pannoramic viewer (3DHISTECH Ltd) was used to scan histology slides and export images (×2 magnification) for image processing and quantifications, which were performed identically for all the samples with the ImageJ software. The TUNEL- and Ki-67-positive tumour areas were thresholded with MaxEntropy and quantified. The ratio of Ki-67-positive (or TUNEL-positive) area to total tumour area was calculated.

**Co-localization analysis**. Pixel-intensity-based Pearson correlation coefficient ($R$) between two channels was calculated using coloc2 plugin (https://imagej.net/Coloc_2) of ImageJ v1.51s with default parameters. Percentage co-localization between the vesicular particles was done using ComDet 0.3.6.1 (https://github.com/ekatrukha/ComDet) plugin of ImageJ v1.51s. Particles were detected in both channels independently at approximated particle sizes of four pixels with sensitivities of signal/noise ratio of 4. Co-localization was determined based on a maximum distance between two particle centres of five pixels and expressed as a percentage.

**Colony formation assay**. Cells, silenced as indicated for 48 h, were plated on six-well plates ($1 \times 10^4$ cells/well) and exposed the following day to different concentrations of the indicated drugs and control cells were treated with DMSO. Drug-containing medium was replenished after 3–4 days and the cells were fixed and stained with 0.2% crystal violet in 10% EtOH for 10 min at RT after 7 days of treatment. Dried plates were then scanned, and the confluency of cells per well was quantified by using the ImageJ (NIH) ColonyArea plug-in[49].

**Statistical analysis**. The GraphPad Prism software and two-tailed Student's t test (paired or unpaired, as appropriate) was used for statistical analysis. Normal

distribution of the data was tested with the Shapiro−Wilk normality test. Unpaired $t$ test was used when normality could not be tested ($n < 8$). When data were not normally distributed, a Mann−Whitney test was used. In proliferation assays, two-way ANOVA was used. $P$ values < 0.1 are shown in graphs.

## Data availability

Breast cancer patient survival data are available from the Kaplan−Meier Plotter in silico biomarker assessment tool (http://kmplot.com) including all RNA expression datasets from 2010, 2012, 2014, and 2017. All the remaining data supporting the findings of this study are available within the paper and its supplementary information files, or from the corresponding author on reasonable request.

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

## Acknowledgements

We thank P. Laasola, J. Siivonen, and S. Collanus for excellent technical assistance, M. Saari for help with the microscopes. The Cell Imaging and Cytometry core facility, University of Turku, Turku Bioscience Centre for technical assistance with imaging. S. Hämälistö for useful suggestions regarding the CAD experiments. H. Hamidi for the scientific illustrations and manuscript editing, J. Westermarck, M. Salmi and the Ivaska lab for critical reading and feedback on the manuscript. This study has been supported by the Academy of Finland (M.P. and J.I.), Academy of Finland CoE for Translational Cancer Research (J.I. and H.J.), an ERC CoG grant 615258 (J.I.), the Sigrid Juselius Foundation, the Orion Research Foundation and the Finnish Cancer Organization (J.I.). P.S. has been supported by the Turku Doctoral Program of Molecular Medicine (TuDMM). E.P. is supported by the Finnish Cultural Foundation.

## Author contributions

Conceptualization, M.P., J.I. and P.S.; Methodology, M.P., J.I., P.S., E.P., P.T., P.B., and O.M.A.; Investigation, M.P., J.I., P.S., E.P., I.P., N.Z.J., A.P., I.S., E.N., H.A.-A., J.L., M.G., H.S., H.J., and M.B.; Writing—original draft, M.P. and J.I.; Writing—review and editing, M.P., J.I., and P.S.; Resources, H.J., P.T., P.B., and J.I.; Funding acquisition, M.P. and J.I.; Supervision, M.P., J.I., and P.T.

## Additional information

**Competing interests:** H.S. and H.J. own stocks of Sartar Therapeutics and are board members. H.J. has a co-appointment at Orion Pharma, and has received fees from Orion Pharma and Neutron Therapeutics Ltd. J.I., M.P., and P.S. have filed a patent application related to these findings. The remaining authors declare no competing interests.

