## [Peer Review File · Nature Communications]

Reviewers' comments:

Reviewer #1 (Remarks to the Author):

Pietilä et al. state that sorLA interacts with HER2, mediates its internalization and recycling to the cell-surface membrane thereby preventing its degradation. Thus, the authors find that the expression of sorLA up-regulate the cellular pool of HER2, HER2 expression on the cell membrane and facilitates HER2 kinase-activity and signaling which eventually promotes cell-proliferation and tumor growth. In contrast, sorLA deficiency is said to generate dysfunctional and CAD-sensitive lysosomes. If this is correct, their study obviously presents important new insight in HER2 trafficking, and offers alternative targets in cancer treatment.

However, despite several studies and theories, HER2 trafficking (and its significance in relation to signaling) is still unclarified. To contribute to clarification rather than adding to the present 'confusion', new theories should therefore be carefully backed up by solid data – and in that respect the present study needs improvement.

1) With regard to the HER2-sorLA interaction: The evidence is based on co-localization and co-precipitation. Yet, the co-localization (which, by the way, is not that impressive) does not in itself prove direct interaction, and the co-precipitations are performed on lysates i.e. they address receptors in solution and not the membrane-attached proteins.

To establish binding between the two (membrane-anchored) receptors cross-linking on whole cells should be performed prior to precipitation. Also, binding experiments using Surface Plasmon Resonance should be carried out to determine the affinity, the involved domains (the possible implications), and the pH-sensitivity of the interaction (as HER2 appears to remain in complex with sorLA in vesicular compartments).

Comments to Fig.1: MDA-MB-361 and BT474 cells (1a) appear to express similar levels of sorLA, why then are the former called intermediate- and the latter high-expressing? In (1e) the MDA-MB-361 lysate contains no HER2 although it can be nicely detected in (1a)? On the other hand, sorLA is hardly visible in lysates of BT474 ('high'-expressing cells)? Only two markers are used in (1b), why then are three (red, green and white) present in JIMT-1 (lower right panel)? Very little GFP-tagged ECD+TM sorLA is precipitated (1g) but it seems to co-precipitate much more (far more than 1:1) HER2 – how can that be?

Finally, C-terminal GFP-tags (1f) are generally not a good choice as they may interfere with the C-terminal sorting-motif and hamper sorting (e.g. Cramer et al. 2010 Traffic).

2) Regarding sorting and proliferation: The authors state that sorLA mediates internalization of HER2 as well as rapid recycling and increase HER2-expression at the surface membrane. In the absence of sorLA they find a significant reduction in total cellular HER2, suggesting that sorLA saves HER2 from lysosomal degradation by transporting it back to the surface membrane. The experimental results however are not convincing (see below for specific comments). Also, it is puzzling that the authors later report (fig. 4a) accumulation of HER2 in sorLA deficient cells (seemingly due to lack of lysosomal degradation), a phenomenon that might increase instead decrease total HER2. If sorLA alters the total HER2-pool, the authors should demonstrate (metabolic labelling, pulse-chase) that it markedly changes the turnover/half-life of HER2. The data presented indicates that the high(er) expression of HER2 on sorLA-cells is accompanied by increased signaling, cell-proliferation and an increase in DCIS tumors after injection of MDA-MB-361 cells into mammary ducts (fig.2 i). Yet, even though the cells after silencing of sorLA do retain ability to proliferate (about 50% compared to sorLA-cells; fig.2h) they appear completely incapable of tumor production, isn't that a bit odd? Could it for instance indicate that while HER2 is important for proliferation, sorLA (not HER2) is decisive for implantation!? It should be remembered that sorLA targets a series of alternative ligands including cytokines, growth factors and transmembrane proteins that might potentially affect proliferation and tumor-implantation. This, and sorLA sorting in general, should be considered and included in the discussion section.

Comments to Fig.2: In (2a) it would be nice if data on untreated cells (and not just shCTRL cell) were shown. Also, how can it be that cell surface HER2 levels (2a) are significantly lower in MDA-MB-361 than in JIMT-1 cells which express little or no sorLA and less HER2 (see fig.1a). According

to data shown in (2b) the 'high-expressing' BT474 cells seem to have a much lower sorLA expression than in Fig. 1a, and the 'medium-expressing' MDA-MB-361 appear to be almost blank? In fig.2 (c,d, and g) like in (2a) results on untreated cells should be included. Concerning (2f) I assume that the shown blots are from separate blots (?), if so they can hardly relate to the same α -tubulin. More importantly, why is there no corresponding results obtained in JMIT-1? Since JMIT-1 cells have little or no sorLA they would be expected to have a 'signaling-pattern' similar to that of BT474-siSorLA# cells, and it would be nice (even crucial) to know if that is the case - and to see if signaling in the JMIT-1 cells can be restored/enhanced by overexpression of wt-sorLA (and perhaps sorLA ECD+TM).

Regarding internalization and recycling: The major pool of sorLA is found in intracellular compartments, and it is well known that sorLA mediates Golgi-endosome transport as well as endocytosis of ligands. Nothing has indicated that the internalized receptor is rapidly recycled. The ligands (unless they are transmembrane and carry their own sorting-signals) are released in endosomes at low pH, and directed to the lysosomes by default. In other words, following endocytosis in (assumed) complex with sorLA, one would expect the two receptors to separate and HER2 to 'continue' on its own like HER2 in the absence of sorLA. Yet, the authors suggest that 100% (!!) of HER2 internalized (by sorLA) in sorLA-GFP cells is recycled (by sorLA) to the surface membrane within 10 min. Unfortunately, the experimental set-ups do not allow these conclusions. To demonstrate that sorLA contributes to the endocytosis of HER2, the authors need to demonstrate that they can block sorLA mediated binding and endocytosis by adding an (excess of) a competing alternative sorLA ligand (e.g. RAP).

The recycling assay is a mystery to me, the description doesn't make sense. In any case, the authors should show (microscopy and WB) how much labelled HER2 that has been internalized after 30 min of endocytosis, and demonstrate that all labeling has returned to the plasma-membrane after 10 min of 'recycling'. To avoid troublesome labelling of other membrane proteins, HER2 could for instance be labelled with Ig (Fab-fragments) instead of biotin - and cross-linking could serve to evidence that internalized HER2 is in complex with sorLA.

Comments to Fig.3: As sorLA is a key issue, co-staining for sorLA should be included in fig.3 (a, c and e). Why such a big difference between intracellular HER2 intensity in shCTRL (3b) and in vehicle (3d)? HER2 appears to accumulate in MDA-MB-361 #3A after 30 min (3g) allegedly due to lack of recycling (or because it is trapped in defective lysosomes?), nevertheless control cells expressing sorLA appear to 'catch up' after 60 min (3f), why??

As mentioned above I am confused by the recycling data (3i). Does 100% recycling mean that after 10 min no biotinylated HER2 is found inside the cell (not that 100% is now seen on the surface membrane)? In other words, does 100% signify that nothing is detected or (preferably!) that 100% of a known internalized amount is now found on the surface membrane??

Regarding lysosome-phenotypes: It is stated that internalized HER2 is slowly degraded and accumulates in large LAMP-1 positive late-endosomes/lysosomes upon sorLA knockdown. Results in MDA-MB-361 may support their notion but again, there are no data on JMIT-1 cells and there should be! It is highly relevant to know if 'low-expressing' JMIT-1 cells accumulate HER2 in defective lysosomes (as would be expected), and if the defect can be restored by overexpression of sorLA.

The same problem concerns the results obtained from DQ red BSA signals and Ebastine sensitivity: Why are there no data on JMIT-1 cells? Also, if intracellular HER2 accumulates in sorLA-deficient cells why then does total HER2 decline as indicated in (2b)?

Comments to Fig. 4: Very large HER2 containing vesicles are shown in the middle and right panel of (4a), but I seems to me that the cells are also twice as big (are you sure it is the same magnification)? In any case I cannot see the same difference in (4b) or in confocal results in BT474 cells (supplementary figure)! A single EM-picture is shown (4d), was 'immature lysosomes' in sorLA expressing and deficient cells quantified by EM? The difference in Ebastine sensitivity between parental MDA-MB-361 and sorLA deficient cells is based on a single point (with practically no sd in either of the two #-cells?) - additional measurements in the critical part of the curves are needed to establish the difference. Finally, what exactly does mean +/- sd indicate (the figure legend is hard to interpret): Is it mean of 4 replicates of one experiment with each cell type, or mean of mean values obtained I three experiments with each cell type, or mean of replicates (3x4)

obtained in three separate experiments with each cell type? Were the experiments performed in parallel?

I abstain from commenting on the bladder cancer results.

Reviewer #2 (Remarks to the Author):

In this very interesting manuscript Pietila and colleagues describe a role for sortilin related receptor 1 (SORLA) in HER2 trafficking and cancer progression. They then go one step further and think about how the effects that SORLA depletion has on HER2 trafficking and lysosome function could be exploited for novel therapeutic approaches. This is, in my opinion, a very promising and novel route that will generate a lot of interest in the wider field.

Importantly, the authors deliver for the first time direct evidence for HER2 internalization and endosomal trafficking. A long overdue correction of early results on HER2 trafficking that have been misinterpreted ever since. The authors summarize this debate in their introduction well. The characterisation of variability of HER2 localisation in several HER2 positive cell lines is very interesting and one of the strengths of this manuscript. SORLA expression levels are shown to influence the localisation and trafficking kinetics of HER2.

One point that is a bit curious in this manuscript is that the correlation of SORLA expression and HER2 levels was not evaluated in primary human breast tissue, but bladder cancer. This IHC is supportive and informative but inclusion of breast samples would have been better aligned with the study. The authors should explain why they omitted breast IHC.

Major points:

-The effect of SORLA on HER2 localisation is very interesting. Is this specific for HER2 or do unrelated receptors behave in a similar way? The authors should include controls, for example integrins, to see if they behave in the same way. Even if other receptors behave analogously, it will not take away from the findings, but help to get closer to an understanding of the effects SORLA has on the endocytic system.

- Measuring trafficking in cell lines with and without SORLA is an interesting approach but the results presented here do not fully convince me that endocytic recycling alone is affected. In Fig3g the kinetics of internalised receptor after SORLA depletion seem to make a sudden jump from 20 to 30 minutes and otherwise behave like wildtype cells. The authors need to calculate rate constant for endocytosis and it would be better to repeat the experiment with addition of a recycling inhibitor like primaquine to assess endocytosis rates in isolation. In addition, recycling rates need to be tested (like in Fig.3i) in MDA-MB-361 to show that SORLA is required for recycling. In Fig.3i effect of SORLA overexpression in JIMT-1 cells needs to be shown in time course to be able to make statement that SORLA promotes HER2 recycling.

- The behaviour of HER2 in lysosomes in SORLA depleted cells need to be characterised better. Knockdown experiments in Fig.4 need to be rescued with SORLA re-expression to show siRNA specificity as performed in Fig.2. The effect on general lysosomal activity in SORLA depleted cells is nicely described in this manuscript, but in order to be able to make a statement about the lysosomal dysfunction influencing HER2 localisation, HER2 turnover itself needs to be investigated. The turnover of HER2 in SORLA depleted cells should be measured with cycloheximide chase or similar to improve the single quantified western blot of HER2 levels after SORLA depletion in Fig. S3a.

Minor:

-The immunoprecipitation of SORLA with HER2 in Fig1e is not very convincing. I recommend the

authors either remove the experiment or include a control with SORLA knockdown.

- Fig.2 F to compare phosphorylation levels it would be helpful to show a stimulated cell sample, the pERK signal looks very strong in this exposure.

Reviewer #3 (Remarks to the Author):

In the present manuscript, the authors investigated the role of Sortilin related receptor 1 (SORLA) in HER2-amplified breast cancer cells. They reported a correlation between SORLA levels and HER2 subcellular localization. In particular, they showed that in SORLA-high/HER2-high expressing cells, HER2 was mainly found at the cell surface, while in SORLA-low/HER2-high cells, HER2 localized also in intracellular compartment (EEA1/VPS35-positive). Starting from this observation, the role of SORLA in HER2 trafficking was investigated, showing that the KD of SORLA in SORLA-high/HER2-high expressing cells caused a reduction of HER2 surface levels (as well as total HER2 levels), and its retention in lamp-1-positive structures, suggesting it accumulated in lysosomes. Overexpression of SORLA in SORLA-low/HER2-high cells increased recycling of HER2 and enhanced its PM level, suggesting that SORLA could be a positive regulator of HER2 recycling. However, the situation is more complex than this, as the KD of SORLA also caused aberrant lysosomal aggregates through an unknown mechanism, and sensitize SORLA-high/HER2-high expressing cells to apoptosis upon treatment with lysosome-accumulating drugs.

The manuscript deals with an unresolved critical issue in HER2-driven cancer cells biology, i.e. the mechanism of HER2 trafficking and its relevance for aberrant HER2 signaling in cancer cells. Despite its relevance, there are several discrepancies among experiments, as well as technical and conceptual issues that make conclusions not enough solid, and that need to be solved prior publication.

1) The authors correlated low SORLA levels in HER2-amplified breast cancer cells with HER2 accumulation in intracellular compartments. However, this correlation is not totally convincing for the following reasons: i) to make their point, authors claim that MDA-MB-361 cells have an intermediate SORLA expression level, but these cells have the same - if not even higher - SORLA level as BT4T74 or HCC119 (see WB in Figure 1), which are considered as high expressing. This is a problem because many of the conclusions reached by the authors are made on the basis of results obtained with these cell model systems; authors should reconcile these incongruences; ii) moreover, even concerning the localization of HER2 in MDA-MB-361, results are also not clear, as, in some experiments, HER2 is mainly at the PM and no major difference are visible between MDA-MB-361 and BT474 (see Fig. 1b); while, in other experiments, HER2 is visualized in intracellular dots colocalizing with EEA1 (Fig. 1c top). These apparent discrepancies reflect the absence of a quantitative and systematic analysis of PM vs. intracellular localization of HER2. To make a point on the correlation between SORLA levels and HER2 localization, authors should provide a quantitative FACS analysis of HER2 localization in the various cell lines used along the study, showing PM vs. cytosolic ratio (e.g. by analyzing the HER2 staining -/+ permeabilization of cells or -/+ acid wash prior fixation of cells).

2) The lack of quantitation of the events shown in the live-TIRF experiment shown in Fig.1d renders it difficult to estimate the relevance of the phenomenon. As this is an initial observation that is further dissected in next figures, in absence of a quantification, I would move the data in the Supplementary results.

3) The co-IP between HER2 and SORLA in MDA-MB-361 is not convincing, as a band at the right MW of SORLA is not visible in the co-IP with HER2 (Fig. 1e left). In both WB (MDA-MB-361 and BT474) a higher exposure where both inputs are visible should be shown in addition to the low exposure, otherwise it is impossible to appreciate if bands in the co-IP are at the expected MW.

4) Authors showed that the KD of SORLA in BT474 (Fig. 2b) decreases not only the surface level of HER2, but also the total protein level (see WB in Fig. 2b, 2f). This is paralleled by an increase in both PM and total protein HER2 level upon SORLA overexpression in JMT-1 and MDA-MB-361 cells. A strong decrease in total HER2 levels was also observed in MDA-MB-361 upon SORLA KD (Supplementary Fig. 2g). However, later on in the manuscript, the authors claim that in MDA-MB-361 they did not observe a major protein decrease upon SORLA KD and they refer to another figure (Supplementary Fig. 3a), which is however in contrast with the previous observation (compare Supplementary Fig. 2g vs. 3a). Authors should reconcile these inconsistencies. From the sum of the data presented, it clearly emerges that SORLA positively regulates not only the PM, but also the total HER2 levels, possibly by skewing the fate of HER2 from a degradative to a recycling fate. Authors should clarify if HER2 is degraded in lysosome or via another mechanism upon SORLA KD, e.g. by treating cells with lysosome inhibitors and see if they can rescue the decrease in protein levels. They should also reconcile the increased HER2 degradation with possible lysosomal defect of these cells. Could it be possible that the excess of HER2 that is not efficiently recycled in SORLA KD cells is partially degraded in the lysosomes, which are however not working properly because overwhelmed by the excess of intracellular HER2? Maybe KD of HER2 in SORLA-KD cells would rescue the lysosomal phenotype.

5) Recycling assay (as in Fig. 3i) should be performed in MDA-MB-361 upon SORLA-KD, to prove that HER2 intracellular accumulation at 30 min of incubation (Fig. 3f and 3g) is due reduced recycling. This would reinforce the idea that SORLA acts by promoting HER2 recycling to the PM.

6) Apparently, the authors did not observe an altered lysosomal localization and function in those cancer cells with HER2 amplification but low SORLA, at variance with what they reported upon SORLA KD in SORLA-high/HER2-high expressing cells (Fig. 4 a-d). Also, accumulation of HER2 in the lysosomes was not reported. If this is the case, authors should comment on this and explain these differences among the different cell systems (along the text or in the discussion).

7) The data on bladder cancer are interesting. However, in order to extend the mechanism proposed by the authors to bladder cancer cells (see Fig.5), authors should show that the KD of SORLA in 5637 cell model system causes HER2 intracellular accumulation and affects recycling, thereby reducing AKT signaling.

Reviewer #1 (Remarks to the Author):

Pietilä et al. state that sorLA interacts with HER2, mediates its internalization and recycling to the cell-surface membrane thereby preventing its degradation. Thus, the authors find that the expression of sorLA up-regulate the cellular pool of HER2, HER2 expression on the cell membrane and facilitates HER2 kinase-activity and signaling which eventually promotes cell-proliferation and tumor growth. In contrast, sorLA deficiency is said to generate dysfunctional and CAD-sensitive lysosomes. If this is correct, their study obviously presents important new insight in HER2 trafficking, and offers alternative targets in cancer treatment.

However, despite several studies and theories, HER2 trafficking (and its significance in relation to signaling) is still unclarified. To contribute to clarification rather than adding to the present ‘confusion’, new theories should therefore be carefully backed up by solid data – and in that respect the present study needs improvement.

We thank the reviewer for these positive and constructive comments and for considering our work to hold potentially important new insights for HER2 trafficking.

1) With regard to the HER2-sorLA interaction: The evidence is based on co-localization and co-precipitation. Yet, the co-localization (which, by the way, is not that impressive) does not in itself prove direct interaction, and the co-precipitations are performed on lysates i.e. they address receptors in solution and not the membrane-attached proteins.

We would like to respectfully point out that we have not claimed interaction in the manuscript. Instead, we have been extremely careful in characterizing the complex using the term associate as indeed we have no evidence for a direct protein-protein interaction between HER2 and SORLA. In fact, we would anticipate that additional factors like an intact plasma membrane, auxiliary proteins or specific post-translational modifications such as phosphorylation or glycosylation may be required for the two proteins to interact.

To establish binding between the two (membrane-anchored) receptors cross-linking on whole cells should be performed prior to precipitation. Also, binding experiments using Surface Plasmon Resonance should be carried out to determine the affinity, the involved domains (the possible implications), and the pH-sensitivity of the interaction (as HER2 appears to remain in complex with sorLA in vesicular compartments).

Even though we are not claiming direct interaction, we have now performed experiments to test whether HER2 and SORLA interact and at least in the various experimental set-ups tested, we have not obtained strong evidence for a direct interaction. We have used purified recombinant SORLA ECD and a commercially obtained HER2-Fc fragment. We have tried: pulldown with His-tagged SORLA ECD bound to His beads with HER2-Fc in solution and, vice versa, HER2-Fc coupled to protein-G beads and incubated with soluble SORLA ECD. Unfortunately, these, and far-western blot assays, give no indication of a direct interaction with the protein fragments tested.

We have also analysed the domain(s) involved. These extensive mapping studies have involved producing and purifying myc and 6xHis-tagged SORLA ECD, and CR-C, BP-EGF and BP-

EGF+CR-C fragments of ECD. Pulldown assays with the recombinant fragments showed that all ECD fragments tested bound to endogenous HER2 from BT474 cell lysate (shown in the new Supplementary Fig. 2b,c). This suggests that the association with HER2 and SORLA occurs via several, potentially weak affinity, direct or indirect interactions (mentioned on page 5 “association with HER2 and SORLA occurs via several, potentially weak affinity, direct or indirect interactions” and in the discussion on page 12 “We found that the ECD of SORLA associates with HER2 (either directly or indirectly).

Comments to Fig.1: MDA-MB-361 and BT474 cells (1a) appear to express similar levels of sorLA, why then are the former called intermediate- and the latter high-expressing?

We appreciate this point and agree that depending on the loading and due to the semi-quantitative nature of WBs, the levels of SORLA on this particular blot may appear similar. We have now investigated thoroughly the levels of SORLA in the MDA-MB-361, BT474 and JIMT-1 cell lines mainly studied in this manuscript. We obtained a new antibody that allowed us to perform quantitative flow cytometry analyses of cell-surface SORLA levels in these cells. We also performed several WB repeats loading the same amount of protein from these cell lines onto the same gels and blotting for total SORLA levels. These new data are shown in new Figures, Fig. 1b and Supplementary Fig. 1b (and below for your convenience). These data indicate that BT474 cells have higher SORLA levels than MDA-MB-361 cells and that the levels of SORLA in JIMT-1 cells are very low.

New Fig. 1b. Quantification of SORLA cell-surface levels by FACS in MDA-MB-361, BT474 and JIMT-1 cells (n=3 from 3 independent experiments).

New Supplementary Fig. 1b. Western blot analysis comparing SORLA and HER2 levels in MDA-MB-361 (n=4), BT474 (n=4) and JIMT-1 (n=3) cells. Values are normalized to MDA-MB-361 levels.

In (1e) the MDA-MB-361 lysate contains no HER2 although it can be nicely detected in (1a)? On the other hand, sorLA is hardly visible in lysates of BT474 (‘high’-expressing cells)?

This point is well taken. These are very different exposures of western blots from the cell lysates and therefore the signal levels cannot be directly compared. However, to provide nicer endogenous IPs we have repeated these experiments and the old IPs are replaced by new ones where SORLA is clearly visible in the BT474 lysate (New Fig 1e; again the absolute band intensities should not be directly compared as they are from separate experiments).

Only two markers are used in (1b), why then are three (red, green and white) present in JIMT-1 (lower right panel)? We apologise for the lack of explanation in the legend. The white indicates co-localisation between the two markers. We mention this now in the legend “Colocalisation of HER2 and EEA1 signal is indicated in white in the merged panels.”

Very little GFP-tagged ECD+TM sorLA is precipitated (1g) but it seems to co-precipitate much more (far more than 1:1) HER2 – how can that be?

The blots of SORLA and HER2 cannot be compared quantitatively as antibody affinities differ in blotting and the exposure shown influences the signal strength. The important comparison is between GFP-SORLA and GFP-tagged ECD+TM SORLA. Both of these are precipitated at similar levels in 1g and the co-precipitate similar levels of HER2, indicating that SORLA-ECD associates with HER2.

Finally, C-terminal GFP-tags (1f) are generally not a good choice as they may interfere with the C-terminal sorting-motif and hamper sorting (e.g. Cramer et al. 2010 Traffic).

We agree that tagging SORLA might interfere with its function. However, we find that our tagged SORLA fully rescues the function of silenced endogenous SORLA in Fig. 2h. Both tagged SORLA and endogenous SORLA have similar endosomal localisation in cells (Fig 1d) and the subcellular distribution of SORLA-GFP (Fig 1d and Supplementary 1c) are very similar to what has been published for untagged exogenously expressed SORLA in other publications, for example J Biol Chem. 2015 Feb 6; 290(6): 3359–3376.

2) Regarding sorting and proliferation: The authors state that sorLA mediates internalization of HER2 as well as rapid recycling and increase HER2-expression at the surface membrane. In the absence of sorLA they find a significant reduction in total cellular HER2, suggesting that sorLA saves HER2 from lysosomal degradation by transporting it back to the surface membrane. The experimental results however are not convincing (see below for specific comments). Also, it is puzzling that the authors later report (fig. 4a) accumulation of HER2 in sorLA deficient cells (seemingly due to lack of lysosomal degradation), a phenomenon that might increase instead decrease total HER2.

We would like to argue that the lysosomal accumulation and the partial HER2 downregulation in SORLA-silenced cells are linked. Endocytosed HER2 is primarily returned to the plasma membrane and does not normally undergo extensive degradation within lysosomes; therefore HER2 has a low degradation rate and we would expect that any increased targeting/accumulation of HER2 to lysosomes, for example in the case of SORLA silencing, would cause a decrease in HER2 levels compared to control.

We show that SORLA silencing correlates with increased HER2 trafficking to lysosomes, which do not appear to be fully functional, therefore HER2 is only partially degraded in these

cells. In summary, given that SORLA silencing has multiple effects: partial lysosome dysfunction (less degradation of lysosome cargo) and increased intracellular HER2 (some of which accumulates in the dysfunctional lysosomes) due to impaired recycling; overall outcome is partial HER2 downregulation and mislocalisation of the receptor.

If sorLA alters the total HER2-pool, the authors should demonstrate (metabolic labelling, pulse-chase) that it markedly changes the turnover/half-life of HER2.

We have observed that the levels of HER2 decline gradually in cells over time such that HER2 downregulation is most evident after 5 days of silencing (or more, see below the data for reviewers) suggesting that this is not an acute response triggered by SORLA loss. However, we have investigated this further as suggested by you and reviewer #2. We have investigated HER2 and SORLA protein stability using a cycloheximide time-course in ctrl and SORLA siRNA cells (as requested by reviewer#2). First, we find that cellular HER2 is remarkably stable. Second, we find that SORLA is very labile with a large fraction of the protein degraded in 8 hours. This unfortunately precludes us from evaluating the role of SORLA in regulating HER2 protein levels in this experimental set-up.

Figure for reviewer: Representative western blot of HER2 protein levels after SORLA silencing at different time points. Quantification of HER2 protein levels from the blots.

Figure for reviewer: Representative western blot of HER2 protein levels in SORLA silenced (36 hours) MDA-MB-361 cells treated with cycloheximide 25 μ g/ml for the indicated time points.

The data presented indicates that the high(er) expression of HER2 on sorLA-cells is accompanied by increased signaling, cell-proliferation and an increase in DCIS tumors after injection of MDA-MB-361 cells into mammary ducts (fig.2 i). Yet, even though the cells after silencing of sorLA do retain ability to proliferate (about 50% compared to sorLA-cells; fig.2h) they appear completely incapable of tumor production, isn't that a bit odd? Could It for instance indicate that while HER2 is important for proliferation, sorLA (not HER2) is decisive for implantation!?

Indeed, the *in vivo* effect of SORLA silencing is more profound in the DCIS model than on the plastic dish *in vitro*. We anticipate this to be due to the fact that when cells are injected intraductally they must feature anoikis-resistance as well as robust proliferation to seed tumors. Furthermore, the *in vivo* environment is different, and most likely more challenging, compared to the *in vitro* conditions that the cells are adapted to. However, we fully agree with the reviewer on the important role of SORLA (potentially via HER2 dependent and independent mechanisms) in this cancer model.

It should be remembered that sorLA targets a series of alternative ligands including cytokines, growth factors and transmembrane proteins that might potentially affect proliferation and tumor-implantation. This, and sorLA sorting in general, should be considered and included in the discussion section.

We fully agree with this important point. It is possible that the requirement for SORLA expression for the oncogenic potential of cancer cells is also linked to regulation of sorting of other cargo in addition to HER2. This is now mentioned in the discussion on page 13 "SORLA could additionally contribute to the oncogenic properties of cancer cells by regulating the trafficking of other cargo".

Comments to Fig.2: In (2a) it would be nice if data on untreated cells (and not just shCRTL cell) were shown.

We would like to respectfully argue that the gold standard is to compare the transfected cells to their relevant control plasmid transfected cells. Mere transfection/plasmid expression may have subtle effects on the cells surface proteins and therefore, with all due respect, we feel that the controls included are the most appropriate.

Also, how can it be that cell surface HER2 levels (2a) are significantly lower in MDA-MB-361 than in JIMT-1 cells which express little or no sorLA and less HER2 (see fig.1a).

The data shown are from separate experiments done over a sequence of a few years using different batches of the HER2 antibody and the absolute values between experiments should not be compared, only the samples within each experiment are comparable. However, we have now done control FACS assays with the cell lines to quantitatively compare the HER2 levels in the cell lines and demonstrate that MDA-MB-361 cells have equal or even slightly higher cell surface HER2 than JIMT-1 cells (New Supplementary Fig. 1e).

New Supplementary Fig. 1e. Flow cytometry analysis of HER2 cell surface levels in BT474, MDA-MB-361, and JIMT-1 cells (n=5 from 4 independent experiments).

According to data shown in (2b) the ‘high-expressing’ BT474 cells seem to have a much lower sorLA expression than in Fig. 1a, and the ‘medium-expressing’ MDA-MB-361 appear to be almost blank?

As mentioned above, different exposures of separate WB experiments cannot be compared, but as suggested, we have compared quantitatively the SORLA levels in these cells (see above, Fig 1b and Supplementary 1b).

In fig.2 (c,d,and g) like in (2a) results on untreated cells should be included.

Please see our response above.

Concerning (2f) I assume that the shown blots are from separate blots (?), if so they can hardly relate to the same a-tubulin.

The reviewer is correct in assuming that all of the shown panels are not from the same membrane. We fully agree with the reviewer on the fact that the shown panel does not belong to the same membrane but are from two different blots. However, the same amount of protein from the same sample was loaded on both gels and the gels were transferred in an identical manner within the same experiment. This was done in order to avoid rounds of membrane stripping that can compromise the blot quality. Moreover, the total AKT was performed as a loading control for the other membrane and the tubulin for the other. Nevertheless, we have now included the tubulin blotted also from the other membrane (please see updated Figure 2f).

More importantly, why is there no corresponding results obtained in JMIT-1? Since JMIT-1 cells have little or no sorLA they would be expected to have a ‘signaling-pattern’ similar to that of BT474-siSorLA# cells, and it would be nice (even crucial) to know if that is the case -

and to see if signaling in the JMIT-1 cells can be restored/enhanced by overexpression of wt-sorLA (and perhaps sorLA ECD+TM).

We agree that if the SORLA-levels were the only difference between these cells lines, this would be a great experiment. However, these cancer cell lines are expected to have multiple other differences as well (please see further justification for this below) and assigning their phenotypic differences to SORLA expression alone would not be well justified. Instead, we show already in this figure that overexpression of SORL-GFP in JIMT-1 cells increases proliferation and cell-surface HER2 levels.

We have also analysed this further to respond to the reviewers question about signalling. We find that overexpression of SORLA in JIMT-1 cells induces an opposite effect in phosphorylation of AKT and 4EBP1 when compared to SORLA silencing in MDA-MB-361 and BT474 cells.

Figure for reviewer: Representative western blot of pAKT S473 and p4EBP1 T37/T46 levels in JIMT-1 GFP control and SORLA-GFP overexpressing cells.

We feel that the following differences between cell lines preclude their direct comparison based on SORLA levels:

Breast cancer, like all other cancers, is a very heterogeneous group of diseases with several subtypes that differ from each other. This is also true even within HER2-amplified breast cancers. HER2-amplified ER+ vs. HER2-amplified ER- breast cancers are very different. Not to mention the differences in transcriptional subgroups (Luminal A, Luminal B, Basal, HER2 enriched). Therefore, it is challenging to compare directly between different cell lines.

The characteristics of JIMT-1 cells is very distinct from BT474 and MDA-MB-361 cells:

<https://cellmodelpassports.sanger.ac.uk/passports/SIDM01037>

Name(s) JIMT-1

Tissue Breast

Cancer Type Breast Carcinoma

Tissue Status Metastasis

Sample Site Pleural effusion

Cancer Type Details Breast Carcinoma

Hormone receptor status and transcriptional subtype: ER-, PR-, Basal (Tanner et al. Mol Cancer Ther. 2004 Dec;3(12):1585-92.)

<https://cellmodelpassports.sanger.ac.uk/passports/SIDM00963>

Name(s) BT-474

Tissue Breast

Cancer Type Breast Carcinoma

Tissue Status Tumour

Sample Site Unknown

Cancer Type Details Invasive Ductal Carcinoma not Otherwise Specified

Hormone receptor status and transcriptional subtype: ER+, PR+, Luminal B (Holliday and Speirs. Breast Cancer Research 2011, **13**:215)

<https://cellmodelpassports.sanger.ac.uk/passports/SIDM00528>

Name(s) MDA-MB-361 (MDA-361)

Tissue Breast

Cancer Type Breast Carcinoma

Tissue Status Metastasis

Sample Site Brain

Cancer Type Details Breast Adenocarcinoma

Hormone receptor status and transcriptional subtype: ER+, PR-, Luminal B (Wang et al. Sci Rep. 2016; 6: 26456.)

Regarding internalization and recycling: The major pool of sorLA is found in intracellular compartments, and it is well known that sorLA mediates Golgi-endosome transport as well as endocytosis of ligands. Nothing has indicated that the internalized receptor is rapidly recycled. The ligands (unless they are transmembrane and carry their own sorting-signals) are released in endosomes at low pH, and directed to the lysosomes by default. In other words, following endocytosis in (assumed) complex with sorLA, one would expect the two receptors to separate and HER2 to 'continue' on its own like HER2 in the absence of sorLA .

We appreciate the reviewer's expertise in the established functions of SORLA in regulating traffic of other cargo proteins in other cell types than breast cancer cells. However, many biological processes are very cell-type and context dependent and assuming that SORLA regulates traffic (of itself and all its cargo) in the same way across different cell types may be misleading.

Yet, the authors suggest that 100% (!) of HER2 internalized (by sorLA) in sorLA-GFP cells is recycled (by sorLA) to the surface membrane within 10 min. Unfortunately, the experimental set-ups do not allow these conclusions.

We are sorry for the lack of clarity in how these data were presented and fully understand the reviewer's confusion. The 100% on y-axis was misleading indeed as it did not refer to the level of recycled HER2 but was normalized to the level of recycled HER2 in the SORLA-GFP cells (which was give the value 100) in comparison to the GFP-ctrl cells. We have now re-analysed the data according to the generally used formula in the field where the loss of biotinylated (=recycled receptor) is normalized to the total (endocytosed after 30 minutes) for each cell type: $(\text{Signaltotal} - \text{Signalremaining}) / \text{Signaltotal}$. The updated figure is included as Fig.3i.

To demonstrate that sorLA contributes to the endocytosis of HER2, the authors need to demonstrate that they can block sorLA mediated binding and endocytosis by adding an (excess of) a competing alternative sorLA ligand (e.g. RAP).

Given that we have no evidence that HER2 association with SORLA is dependent on a binding site that would be overlapping with known SORLA ligands, we are afraid this might not be a feasible experiment within the scope of this manuscript.

The recycling assay is a mystery to me, the description doesn't make sense. (please see explanation above) In any case, the authors should show (microscopy and WB) how much labelled HER2 that has been internalized after 30 min of endocytosis, and demonstrate that all labeling has returned to the plasma-membrane after 10 min of 'recycling'. To avoid troublesome labelling of other membrane proteins, HER2 could for instance be labelled with Ig (Fab-fragments) instead of biotin – and cross-linking could serve to evidence that internalized HER2 is in complex with sorLA.

We have now performed additional experiments as suggested by the reviewer: labelling cell-surface HER2 with antibody on ice, allowing trafficking for 45 minutes, performing an acid-wash (removing any remaining cell-surface antibody) and allowing recycling for 30 minutes. These have been imaged with microscopy, as suggested, and the signal of HER2 antibody on the cell-surface or inside the cell has been quantified. These new data with ctrl and siSORLA MDA-MB-361 demonstrate that SORLA silencing significantly impairs HER2 recycling to the plasma membrane (New Fig. 3g and below). In addition, we have validated the recycling data with the JIMT-1 cells using cell-surface biotinylation (biotin-based endocytosis assays are frequently used in the field by many groups) and an ELISA assay detecting the biotinylation of the captured HER2 (New Fig Supplementary 4a and below).

New Figure 3g: Microscopy-based HER2 recycling assay. Control or SORLA siRNA treated MDA-MB-361 cells were labelled with anti-HER2 IgG on ice followed by 45 min of endocytosis at +37 C, subsequent removal of IgG on cell surface by acid wash and a recycling step at +37 C for 30 min. The cells were fixed, labelled with anti-human secondary antibody and imaged with a confocal microscope. n=26-57 cells/condition from two independent experiments. Values of individual cells are plotted and median+/- interquartile range is displayed. Non-parametric Kruskal-Wallis test was used.

New Fig Supplementary 4a: Recycling of HER2 at 10 min after 30 min of endocytosis. Two antibodies used: red points (herceptin), black points (c-erbB-2 A0485). JIMT-1 cells using cell surface biotinylation and an ELISA assay detecting the biotinylation of the captured HER2.

Comments to Fig.3: As sorLA is a key issue, co-staining for sorLA should be included in fig.3 (a, c and e).

We would like to argue that both SORLA and HER2 are equally important for the story and furthermore, as these experiments are aimed to characterize the unknown in HER2 traffic, repeating all these experiment to be able to add the SORLA staining would not bring significant added-value to the conclusions drawn from these experiments.

Why such a big difference between intracellular HER2 intensity in shCRTL (3b) and in vehicle (3d)?

These stainings and quantifications are performed by two different people with two different microscopes (spinning disc and LSM780) with different microscope settings. Accordingly, the absolute intensity values between these two experiments will vary just because of the different settings/microscopes/persons running the experiment. Moreover, the purpose is not to compare between the two different experiments, but within the experiments that were imaged identically at the same time.

HER2 appears to accumulate in MDA-MB-361 #3A after 30 min (3g) allegedly due to lack of recycling (or because it is trapped in defective lysosomes?), nevertheless control cells expressing sorLA appear to ‘catch up’ after 60 min (3f), why??

This is an important point and we are grateful to the reviewer for pointing this out. Prompted by this criticism, we have repeated these HER2 endocytosis assays (that were based on two biological replicates) in the MDA-MB-361 cells using the somewhat more quantitative capture ELISA system and based on these more thorough analyses there seems to be no statistically significant difference in HER2 endocytosis even at 30 minutes. We have therefore removed these data and replaced them with the recycling data shown above.

As mentioned above I am confused by the recycling data (3i). Does 100% recycling mean that after 10 min no biotinylated HER2 is found inside the cell (not that 100% is now seen on the surface membrane)? In other words, does 100% signify that nothing is detected or (preferably!) that 100% of a known internalized amount is now found on the surface membrane??

We are sorry for the lack of clarity in how these data were presented and fully understand the reviewer's confusion. The 100% on y-axis was misleading indeed as it did not refer to the level of recycled HER2 but was normalized to the level of recycled HER2 in the SORLA-GFP cells (which was give the value 100) in comparison to the GFP-ctrl cells. We have now re-analysed the data according to the generally used formula in the field where the loss of biotinylated (=recycled receptor) is normalized to the total (endocytosed after 30 minutes) for each cell type: $(\text{Signal}_{\text{total}} - \text{Signal}_{\text{remaining}}) / \text{Signal}_{\text{total}}$. The updated figure is included as Fig.3i

Regarding lysosome-phenotypes: It is stated that internalized HER2 is slowly degraded and accumulates in large LAMP-1 positive late-endosomes/lysosomes upon sorLA knockdown. Results in MDA-MB-361 may support their notion but again, there are no data on JMIT-1 cells and there should be! It is highly relevant to know if 'low-expressing' JMIT-1 cells accumulate HER2 in defective lysosomes (as would be expected), and if the defect can be restored by overexpression of sorLA. The same problem concerns the results obtained from DQ red BSA signals and Ebastine sensitivity: Why are there no data on JMIT-1 cells?

Figure for reviewer: Staining of LAMP1 in GFP and SORLA-GFP expressing JMIT-1 cells.

As explained above we do not think that these two cell types can be compared directly merely based on their different SORLA levels as there are many other biological differences between

these cells. However, for the benefit of the reviewer, we have stained lysosomes in JIMT-1 cells and we do not detect obviously enlarged lysosomes in these cells basally (see figure above).

Also, if intracellular HER2 accumulates in sorLA-deficient cells why then does total HER2 decline as indicated in (2b)? As the DQ-BSA data demonstrate, the lysosomes are only partially defective and therefore, most likely slowly facilitating turnover of HER2 mounting to lower HER2 levels over several days as shown in Fig. 2b.

Comments to Fig. 4: Very large HER2 containing vesicles are shown in the middle and right panel of (4a), but I seems to me that the cells are also twice as big (are you sure it is the same magnification)?

We realise that the cell borders are hard to envision when the cell surface HER2 signal is lower in the SORLA silenced cells. The magnification used to image the cells is the same. To confirm the same magnification, please find attached DAPI staining of the cells in Figure 4a (Figure for reviewer below, top). Also the cell area was quantified to show that the SORLA silencing led to modest increase in cell area (Figure for reviewer below, bottom), but still indicating that samples were imaged with the same magnification.

Figure for reviewer. DAPI staining of the same cells in Figure 4a panel indicating similar nuclear size that confirms the same magnification used when cells were imaged.

Figure for reviewer. Quantification of cell area (pixels) of the cells indicating modest increase in cell area of SORLA silenced cells when compared to shCTRL cells, however, the difference is so small that it can't be explained by different magnification.

In any case I cannot see the same difference in (4b) or in confocal results in BT474 cells (supplementary figure)!

We realise that due to different magnification, the lysosomes may look different in the figures. However, we have provided careful quantification of increased lysosome aggregation in shSORLA cells both for 361 cells (Fig 4c) and BT474 Fig Supplementary 5c.

A single EM-picture is shown (4d), was 'immature lysosomes' in sorLA expressing and deficient cells quantified by EM?

We have now quantified the lysosome phenotypes from EM images. We classified the lysosomes into two groups by morphology, dark, homogenous and smaller than 1 µm and lighter, multivesicular and larger than 1 µm and scored the number of structures falling into these two categories from the two cells types. The results (shown below) validate that there are significantly more large lysosomes in the SORLA silenced cells.

Figure for reviewer:

The difference in Ebastine sensitivity between parental MDA-MB-361 and sorLA deficient cells is based on a single point (with practically no sd in either of the two #-cells?) – additional measurements in the critical part of the curves are needed to establish the difference. Finally, what exactly does mean \pm sd indicate (the figure legend is hard to interpret): Is it mean of 4 replicates of one experiment with each cell type, or mean of mean values obtained in three experiments with each cell type, or mean of replicates (3x4) obtained in three separate experiments with each cell type? Were the experiments performed in parallel?

The reason why the SD is not visible in that one concentration is because of very small variation of results on that point (basically very small SD). The data are from three completely independent experiments (silencing, plating, treatment etc. performed at different times, week/s apart). In each of the independent experiments we had 4 technical replicates. On the plot, the n=12 represents all the 4 technical replicates from the 3 independent experiments.

We would also like to point out that we performed and confirmed the sensitivity of silenced cells to ebastine on Figure 4h-i (now panels 4g and 4h) by using the ebastine concentration that is within the window of higher sensitivity of silenced cells and detected increased sensitivity and induction of apoptosis. These data confirm the results from the previous toxicity curves. We do agree that in the MDA-MB-361 cells the sensitivity is not dramatically higher over a wide range of different ebastine concentrations, but there is clearly a window of higher sensitivity that can be utilized to trigger apoptosis specifically in the silenced cells (old Fig. 4h-i, now panels 4g and 4h).

Reviewer #2 (Remarks to the Author):

In this very interesting manuscript Pietila and colleagues describe a role for sortilin related receptor 1 (SORLA) in HER2 trafficking and cancer progression. They then go one step further and think about how the effects that SORLA depletion has on HER2 trafficking and lysosome function could be exploited for novel therapeutic approaches. This is, in my opinion, a very promising and novel route that will generate a lot of interest in the wider field.

Importantly, the authors deliver for the first time direct evidence for HER2 internalization and endosomal trafficking. A long overdue correction of early results on HER2 trafficking that have been misinterpreted ever since. The authors summarize this debate in their introduction well. The characterisation of variability of HER2 localisation in several HER2 positive cell lines is very interesting and one of the strengths of this manuscript. SORLA expression levels are shown to influence the localisation and trafficking kinetics of HER2.

We thank the reviewer for these positive, encouraging and carefully considered comments.

One point that is a bit curious in this manuscript is that the correlation of SORLA expression and HER2 levels was not evaluated in primary human breast tissue, but bladder cancer. This IHC is supportive and informative but inclusion of breast samples would have been better aligned with the study. The authors should explain why they omitted breast IHC.

We have collaborated with two clinicians and performed IHC analysis of a very large set of breast tissue (FinHer TMA series; containing high risk BrCa tumours and Herceptin-treated HER2+ tumours). These new data indicate that in this patient cohort there were both SORLA-low (62% of the 199 HER2-amplified tumours) and SORLA-high (38% of the HER2-amplified specimens) breast cancers (New Fig 5a). This is a little unexpected and at the moment we do not have any clear explanation why the SORL/HER2 correlation observed in the cell lines and the bladder cancer specimens is not evident in all the HER2+ breast cancer samples. We further followed this up by using the Km-blotter biomarker assessment tool. These data show that high *SORL1* gene expression correlates significantly with poor patient outcome specifically in HER2+ breast cancers (New Fig 5b). Thus, SORLA may have a prognostic value in HER2-amplified breast cancers.

Major points:

-The effect of SORLA on HER2 localisation is very interesting. Is this specific for HER2 or do unrelated receptors behave in a similar way? The authors should include controls, for example integrins, to see if they behave in the same way. Even if other receptors behave analogously, it will not take away from the findings, but help to get closer to an understanding of the effects SORLA has on the endocytic system.

We would like to thank the reviewer for this suggestion. As we find that SORLA mainly facilitates the cell-surface levels and recycling of HER2, we have analysed these aspects also for integrins, as suggested. FACS analysis indicates a decrease in cell-surface ITGB1 levels (New Supplementary Fig 3b and below).

New Supplementary Fig. 3b

(b) Flow cytometry analysis of ITGB1 levels on the cell surface (detected with P5D2 antibody) in BT474 cells after silencing with control (siCTRL) and SORLA (siSORLA #3 and #4) siRNAs (mean \pm s.d of n = 3 independent experiments; statistical analysis: unpaired Student's t-test).

- Measuring trafficking in cell lines with and without SORLA is an interesting approach but the results presented here do not fully convince me that endocytic recycling alone is affected. In Fig3g the kinetics of internalised receptor after SORLA depletion seem to make a sudden jump from 20 to 30 minutes and otherwise behave like wildtype cells. The authors need to calculate rate constant for endocytosis and it would be better to repeat the experiment with addition of a recycling inhibitor like primaquine to assess endocytosis rates in isolation.

This is an important point and we are grateful to the reviewer for pointing this out. Prompted by this criticism, we have repeated these HER2 endocytosis assays (that were based on two biological replicates) in the MDA-MB-361 cells using the somewhat more quantitative capture ELISA system and based on these more thorough analyses there seems to be no statistically significant difference in HER2 endocytosis even at 30 minutes. We have therefore removed these data and replaced them with our new recycling experiments (see below), which indicate that SORLA silencing significantly inhibits recycling of HER2 back to the plasma membrane in MDA-MB-361 cells.

In addition, recycling rates need to be tested (like in Fig.3i) in MDA-MB-361 to show that SORLA is required for recycling. In Fig.3i effect of SORLA overexpression in JIMT-1 cells needs to be shown in time course to be able to make statement that SORLA promotes HER2 recycling.

All three reviewers considered this to be an important point and requested new experiments with different specific approaches. Thus, to respond to these requests in the best possible and comprehensive manner, we have performed an imaging based recycling assay in the SORLA silenced MDA-MB-361 cells (New Fig. 3g and below). We labelled cell surface HER2 with antibody on ice, allowed trafficking for 45 minutes, acid washed any remaining cell-surface antibody and allowed recycling for 30 minutes. The cells were imaged with a microscope and

the signal of HER2 antibody on the cell surface or inside the cell was quantified. These new data with ctrl and siSORLA MDA-MB-361 cells demonstrate that SORLA silencing significantly impairs HER2 recycling to the plasma membrane (New Fig. 3g).

New Fig. 3g Microscopy-based HER2 recycling assay. Control or SORLA siRNA treated MDA-MB-361 cells were labeled with anti-HER2 IgG on ice followed by 45min endocytosis at +37C, subsequent removal of IgG on cell surface by acid wash and a recycling step again at 37C for 30 min. The cells were fixed, labelled with anti-human secondary antibody and imaged with a confocal microscope. n=26-57 cells/condition from two independent experiments. Values of individual cells are plotted and median+/- interquartile range is displayed. Non-parametric Kruskal-Wallis test was used.

We have also made a time course of the JIMT-1 recycling assay using the capture ELISA system (New Supplementary Fig 4a and below). These data are in line with the immunoprecipitation based recycling assay (Fig. 3i) in the JIMT-1 cells. Overexpression of SORLA increases recycling of HER2 to the plasma membrane in 10 minutes. However, at 15 minutes the difference between JIMT-1 GFP and GFP-SORLA cells is lost or even slightly inverted. This could be due to rapid re-endocytosis of HER2 in the GFP-SORLA cells (resulting in the recycled biotinylated HER2 becoming again protected from biotin cleavage – in this assay this will be indistinguishable from the non-recycled pool). In addition, we have tested ITGB1 traffic (as requested by you above). Similarly to HER2, SORLA silencing in MDA-MB-361 cells does not influence ITGB1 endocytosis but significantly attenuates ITGB1 recycling (New Supplementary Fig. 4b, c and below)

Supplemental Figure 4. (a) ELISA immunoassay based analysis of biotin-labelled cell surface HER2 recycling at 5 min, 10 min and 15 min (30 min endocytosis) in JIMT-1 cells with two different capture antibodies: red points (trastuzumab), black points (c-erbB-2 A0485). Bars represents average HER2 detected with the two antibodies. **b)** ELISA immunoassay based analysis of biotin-labelled cell-surface β 1 integrin (9EG7) internalization after 15 min and 30 min in MDA-MB-361 cells treated with scramble (siCTRL) and SORLA (siSORLA) siRNAs for 72 h (data are mean + SD; n = 3 independent experiments; statistical analysis: unpaired Student's t-test) **c)** ELISA immunoassay based analysis of biotin-labelled cell-surface β 1 integrin recycling after 10 min (30 min endocytosis) in MDA-MB-361 cells treated with scramble (siCTRL) and SORLA (siSORLA) siRNAs for 72 h (data are mean + SD; n = 3 independent experiments; statistical analysis: unpaired Student's t-test).

- The behaviour of HER2 in lysosomes in SORLA depleted cells need to be characterised better. Knockdown experiments in Fig.4 need to be rescued with SORLA re-expression to show siRNA specificity as performed in Fig.2.

We would like to respectfully emphasize that these experiments have already been controlled for off-target effects by using up to 4 independent siRNAs in 2 different cell lines (former Supplementary Fig 3d now Supplementary 5d). Therefore, even though we agree that a rescue would validate the specificity even further, we feel it would not have significantly increased the credibility of these data.

The effect on general lysosomal activity in SORLA depleted cells is nicely described in this manuscript, but in order to be able to make a statement about the lysosomal dysfunction influencing HER2 localisation, HER2 turnover itself needs to be investigated. The turnover of HER2 in SORLA depleted cells should be measured with cycloheximide chase or similar to improve the single quantified western blot of HER2 levels after SORLA depletion in Fig. S3a. We would like to respectfully point out to the reviewer that in addition to the HER2 quantification shown in Supplementary Fig 5a, we had also analysed HER2 levels carefully in SORLA silenced BT474 cells and SORLA-GFP overexpressing MDA-MB-361 and JIMT-1 cells (Fig 2c and below for convenience). These data show that HER2 protein levels are significantly reduced by SORLA silencing and promoted by SORLA overexpression. However, we have investigated HER2 and SORLA protein stability using a cycloheximide time-course in ctrl and SORLA siRNA cells as suggested. First, we find that cellular HER2 is remarkably stable. Second, we find that SORLA is very labile with a large fraction of the protein degraded in 8 hours. This unfortunately precludes us from evaluating the role of SORLA in regulating HER2 protein levels in this experimental set-up.

Figure 2c: (c) Quantification of total HER2 protein levels in the indicated transfectants (mean \pm s.d of n = 3 independent experiments; statistical analysis: unpaired Student's t-test).

Figure for reviewer: Representative western blot of HER2 protein levels in SORLA silenced (36 hours) MDA-MB-361 cells treated with cycloheximide 25 μ g/ml for the indicated time points.

Minor:

-The immunoprecipitation of SORLA with HER2 in Fig1e is not very convincing. I recommend the authors either remove the experiment or include a control with SORLA knockdown.

We have now repeated the IPs in Fig 1e and replaced them with more representative ones (New Fig 1e). In addition, we have performed the suggested IP with silencing and these data are included below for the reviewer.

- Fig.2 F to compare phosphorylation levels it would be helpful to show a stimulated cell sample, the pERK signal looks very strong in this exposure.

We totally agree that comparing the signalling between SORLA silenced and control cells should be done in more controlled conditions, since phosphorylation of AKT and many other signalling components fluctuate depending on cell cycle, growth phase, confluency etc. We have also compared the pAKT levels after serum starvation and re-activation by addition of serum in SORLA silenced and control cells. Even though the phosphorylation of AKT in this case may not be completely dependent on HER2 activity, we do see a slower and attenuated phosphorylation of AKT in silenced cells after starvation and re-activation.

Figure for reviewer: Control and SORLA silenced (siSORLA #3) cells were first starved by growing cells without serum for 4 hours. After re-addition of serum-containing medium, cells were harvested at different time points. pAKT S473 was detected by western blot to monitor AKT activation. Results are mean \pm s.d. from two independent experiments. Representative western blot of pAKT S473 levels after starvation and re-activation.

Reviewer #3 (Remarks to the Author):

In the present manuscript, the authors investigated the role of Sortilin related receptor 1 (SORLA) in HER2-amplified breast cancer cells. They reported a correlation between SORLA levels and HER2 subcellular localization. In particular, they showed that in SORLA-high/HER2-high expressing cells, HER2 was mainly found at the cell surface, while in SORLA-low/HER2-high cells, HER2 localized also in intracellular compartment (EEA1/VPS35-positive). Starting from this observation, the role of SORLA in HER2 trafficking was investigated, showing that the KD of SORLA in SORLA-high/HER2-high expressing cells caused a reduction of HER2 surface levels (as well as total HER2 levels), and its retention in lamp-1-positive structures, suggesting it accumulated in lysosomes. Overexpression of SORLA in SORLA-low/HER2-high cells increased recycling of HER2 and enhanced its PM level, suggesting that SORLA could be a positive regulator of HER2 recycling. However, the situation is more complex than this, as the KD of SORLA also caused aberrant lysosomal aggregates through an unknown mechanism, and sensitize SORLA-high/HER2-high expressing cells to apoptosis upon treatment with lysosome-accumulating drugs.

The manuscript deals with an unresolved critical issue in HER2-driven cancer cells biology, i.e. the mechanism of HER2 trafficking and its relevance for aberrant HER2 signaling in cancer cells. Despite its relevance, there are several discrepancies among experiments, as well as technical and conceptual issues that make conclusions not enough solid, and that need to be solved prior publication.

We are grateful to the reviewer for these encouraging comments and the extremely useful suggestions that have helped us to further strengthen the manuscript.

1) The authors correlated low SORLA levels in HER2-amplified breast cancer cells with HER2 accumulation in intracellular compartments. However, this correlation is not totally convincing for the following reasons: i) to make their point, authors claim that MDA-MB-361 cells have an intermediate SORLA expression level, but these cells have the same - if not even higher - SORLA level as BT4T74 or HCC119 (see WB in Figure 1), which are considered as high expressing. This is a problem because many of the conclusions reached by the authors are made on the basis of results obtained with these cell model systems; authors should reconcile these incongruences;

We appreciate this point and agree that depending on the loading and due to the semi-quantitative nature of WBs, the levels of SORLA on this particular blot may appear similar. We have now investigated thoroughly the levels of SORLA in the MDA-MB-361, BT474 and JIMT-1 cell lines mainly studied in this manuscript. We obtained a new antibody that allowed us to perform quantitative flow cytometry analyses of cell-surface SORLA levels in these cells. We also performed several WB repeats loading the same amount of protein from these cell lines onto the same gels and blotting for total SORLA levels. This is now mentioned in the text on page 3, and shown in new Figs 1b and S1b). These data indicate that BT474 have higher SORLA levels than MDA-MB-361 cells and that the levels of SORLA in JIMT-1 cells is very low.

New Fig. 1b. Quantification of SORLA cell surface levels by FACS in MDA-MB-361, BT474 and JIMT-1 cells (n=3 from 3 independent experiments).

New Supplementary Figure 1b. Western blot analysis comparing SORLA and HER2 levels in MDA-MB-361 (n=4), BT474 (n=4) and JIMT-1 (n=3) cells. Values are normalized to MDA-MB-361 levels.

ii) moreover, even concerning the localization of HER2 in MDA-MB-361, results are also not clear, as, in some experiments, HER2 is mainly at the PM and no major difference are visible between MDA-MB-361 and BT474 (see Fig. 1b); while, in other experiments, HER2 is visualized in intracellular dots colocalizing with EEA1 (Fig. 1c top).

These apparent discrepancies reflect the absence of a quantitative and systematic analysis of PM vs. intracellular localization of HER2. To make a point on the correlation between SORLA levels and HER2 localization, authors should provide a quantitative FACS analysis of HER2 localization in the various cell lines used along the study, showing PM vs. cytosolic ratio (e.g. by analyzing the HER2 staining -/+ permeabilization of cells or -/+ acid wash prior fixation of cells). We fully agree on this criticism. First, we have replaced the micrographs in Fig. 1b (Now panel 1c in the revised manuscript and below for your convenience) with a more representative ones to visualize more clearly the plasma membrane and the intracellular HER2.

In addition, we have now performed the suggested quantitative FACS assays on HER2 localization suggested by the reviewer. In line with the data included in the manuscript we find that in BT474 cells HER2 labelling +/- cell permeabilization is almost identical. In MDA-MB-361 and JIMT-1 cells the HER2 signal from permeabilized samples is higher, than the signal from non-permeabilized samples, indicating that in these cells a sub-fraction of the HER2 is intracellular. However, it is important to note that the permeabilization procedure dissolves parts of the plasma membrane thus reducing somewhat the signal from the surface receptors in permeabilized samples. Therefore, these data do not allow us to conclusively define the PM vs cytosolic ratio of HER2 but rather demonstrate that BT474 cells appear to have higher HER2 cell-surface localization than MDA-MB-361 and JIMT-1 cells. Intracellular tubulin was labelled to confirm successful cell permeabilization.

Figure for Reviewer: HER2 and tubulin labelling in non-permeabilized and permeabilized (0.5% saponin 0.5% BSA in PBS for 10 min at RT) BT474, MDA-MB-361 and JIMT-1 cells (HER2 n=5 from 4 independent experiments; tubulin n=1).

2) The lack of quantitation of the events shown in the live-TIRF experiment shown in Fig. 1d renders it difficult to estimate the relevance of the phenomenon. As this is an initial observation that is further dissected in next figures, in absence of a quantification, I would move the data in the Supplementary results.

These data have been moved to Supplementary Fig 1g as suggested.

3) The co-IP between HER2 and SORLA in MDA-MB-361 is not convincing, as a band at the right MW of SORLA is not visible in the co-IP with HER2 (Fig. 1e left). In both WB (MDA-MB-361 and BT474) a higher exposure where both inputs are visible should be shown in addition to the low exposure, otherwise it is impossible to appreciate if bands in the co-IP are at the expected MW.

We have now repeated the IPs in Fig 1e and replaced them with more representative ones (New Fig 1e).

4) Authors showed that the KD of SORLA in BT474 (Fig. 2b) decreases not only the surface level of HER2, but also the total protein level (see WB in Fig. 2b, 2f). This is paralleled by an increase in both PM and total protein HER2 level upon SORLA overexpression in JMT-1 and MDA-MB-361 cells. A strong decrease in total HER2 levels was also observed in MDA-MB-361 upon SORLA KD (Supplementary Fig. 2g). However, later on in the manuscript, the authors claim that in MDA-MB-361 they did not observe a major protein decrease upon SORLA KD and they refer to another figure (Supplementary Fig. 3a), which is however in contrast with the previous observation (compare Supplementary Fig. 2g vs. 3a). Authors should reconcile these inconsistencies.

We would like to thank the reviewer for carefully reviewing our data and fully appreciate this point. For reasons that we do not currently fully understand, the extent of HER2 downregulation is variable between experiments. HER2 downregulation on the cell surface and HER2 total levels are often more pronounced in the BT474 cells whereas in MDA-MB-361 cells this varies from modest (20% reduction) to much more dramatic downregulation. To illustrate this point we have included below another siSORLA western blot that is not part of the manuscript showing again a much more dramatic HER2 downregulation (which is in line with the changes shown in the manuscript in supplementary figure 2g (now Supplementary Fig. 3h in the revised manuscript)). We would like to emphasize, however, that even though the magnitude of HER2 downregulation varies in MDA-MB-361 cells between experiments for an unknown reason, the downregulation is observed consistently in all experiments.

In addition, the KD shSORLA MDA-MB-361 and BT474 cells cannot be maintained for long, since the cells do not proliferate well and appear stressed. This could lead to the situation that the cells in which HER2 levels are reducing dramatically are selected out and the ones that still express HER2 in some extent are selected. Moreover, generation of the shSORLA cells with lentivirus takes some time before we are able to characterize them, so there is a chance that some of the cells in which HER2 levels are drastically reduced are already lost.

From the sum of the data presented, it clearly emerges that SORLA positively regulates not only the PM, but also the total HER2 levels, possibly by skewing the fate of HER2 from a degradative to a recycling fate. Authors should clarify if HER2 is degraded in lysosome or via another mechanism upon SORLA KD, e.g. by treating cells with lysosome inhibitors and see if they can rescue the decrease in protein levels.

We fully agree with the reviewer on the importance of highlighting the HER2 degradation pathway. According to several published reports and our unpublished observations, HER2 is remarkably stable in BrCa cells, suggesting that only a small subset of the receptor traffics to a degradative compartment/complex in cells. This is presumably due to the low endocytosis/rapid recycling of the receptor. However, to investigate this further, as suggested, we individually inhibited the lysosome and proteasome in both control and SORLA-silenced cells. On the one hand, in control cells the lysosome acidification inhibitor bafilomycin A1 (30 nM, 3h) increases HER2 levels by about 20% while the proteasome inhibitor MG132 has no such effect. This indicates that some HER2 is degraded through the lysosomal pathway. On the other hand, no significant effect was obtained upon lysosome inhibition of SORLA-silenced cells (n=3). This is most likely due to the overall reduced lysosomal activity of the SORLA silenced cells (as shown with DQ-BSA degradation assays in the manuscript). Therefore, even though a bigger proportion of HER2 is in lysosomes in SORLA silenced cells, this pool is turned over slowly due to the attenuated lysosome activity. We did not perform longer Bafilomycin A1 treatment time points due to non-specific cell cytotoxicity.

Figure for reviewer: control and SORLA-depleted MDA-MB-361 cells were treated with bafilomycin A1 (30nM) or MG132 (10µM) for 3h. Whole cell lysates were then blotted with the indicated antibodies. All values represent means \pm SD of 3 biological replicates. Student's t-test.

They should also reconcile the increased HER2 degradation with possible lysosomal defect of these cells. Could it be possible that the excess of HER2 that is not efficiently recycled in SORLA KD cells is partially degraded in the lysosomes, which are however not working properly because overwhelmed by the excess of intracellular HER2? Maybe KD of HER2 in SORLA-KD cells would rescue the lysosomal phenotype.

This was a great suggestion. We have now performed the suggested experiment by combining SORLA and HER2 silencing. Very intriguingly, silencing of HER2 significantly attenuates the lysosome aggregation phenotype, indicating that this could indeed be linked to the lysosomes being overwhelmed. These new data are shown as new figure 4d and below for your convenience) and discussed in the text on page 8: “Interestingly, this is linked to the altered traffic of HER2. Silencing of HER2 in conjunction with SORLA reduced lysosomal aggregation significantly (Fig 4d). This suggests that the excess of HER2 that is not efficiently recycled in SORLA silenced cells is overwhelming the lysosomes contributing to their compromised function”.

Immunofluorescence imaging and quantification of lysosomal aggregation in MDA-MB-361 cells after scramble (siCTRL) or SORLA (siSORLA#3, #4) or SORLA and HER2 siRNA silencing (images taken with Zeiss LSM880 63x objective. For the quantification, raw images from the middle plane of the cells were filtered with median filter (1.5 pixel radius), turned binary and segmented using watershed segmentation).

Data from two independent experiments are displayed as box plots, n = 121-182 cells per condition; statistical analysis: Mann-Whitney test.

5) Recycling assay (as in Fig. 3i) should be performed in MDA-MB-361 upon SORLA-KD, to prove that HER2 intracellular accumulation at 30 min of incubation (Fig. 3f) is due reduced recycling. This would reinforce the idea that SORLA acts by promoting HER2 recycling to the PM.

This is an important point and we are grateful to the reviewer for suggesting this. We have performed the requested recycling experiment (see below) in MDA-MB-361 cells and the data indicate that SORLA silencing significantly inhibits recycling of HER2 back to the plasma membrane.

We labelled cell surface HER2 with antibody on ice, allowed trafficking for 45 minutes, acid washed any remaining cell-surface antibody and allowed recycling for 30 minutes. The cells were imaged with a microscope and the signal of HER2 antibody on the cell surface or inside the cell was quantified. These new data with ctrl and siSORLA MDA-MB-361 cells demonstrate that SORLA silencing significantly impairs HER2 recycling to the plasma membrane (New Fig. 3g).

New Fig. 3g Microscopy-based HER2 recycling assay. Control or SORLA siRNA treated MDA-MB-361 cells were labelled with anti-HER2 IgG on ice followed by 45 min endocytosis at +37C, subsequent removal of IgG on cell surface by acid wash and a recycling step again at 37C for 30 min. The cells were fixed, labelled with anti-human secondary antibody and imaged

with confocal microscope. n=26-57 cells/condition from two independent experiments. Values of individual cells are plotted and median+/- interquartile range is displayed. Non-parametric Kruskal-Wallis test was used.

6) Apparently, the authors did not observe an altered lysosomal localization and function in those cancer cells with HER2 amplification but low SORLA, at variance with what they reported upon SORLA KD in SORLA-high/HER2-high expressing cells (Fig. 4 a-d). Also, accumulation of HER2 in the lysosomes was not reported. If this is the case, authors should comment on this and explain these differences among the different cell systems (along the text or in the discussion).

We agree that if the SORLA levels was the only distinction between these cell lines, this would be a great experiment to do. However, The SORLA-low JIMT-1 cells are also biologically different to the intermediated MDA-MB-361 and the SORLA-high BT474 cells in many other respects. Breast cancer like all other cancers is a very heterogeneous group of disease with several subtypes that differ from each other. This is also true even within HER2-amplified breast cancers. HER2-amplified ER+ vs. HER2-amplified ER- breast cancers are very different, not to mention the differences in transcriptional subgroups (Luminal A, Luminal B, Basal, HER2 enriched). Therefore, it is challenging to compare directly between different cell lines.

The characteristics of JIMT-1 is very distinct from BT474 and MDA-MB-361:

<https://cellmodelpassports.sanger.ac.uk/passports/SIDM01037>

Name(s) JIMT-1

Tissue Breast

Cancer Type Breast Carcinoma

Tissue Status Metastasis

Sample Site Pleural effusion

Cancer Type Details Breast Carcinoma

Hormone receptor status and transcriptional subtype: ER-, PR-, Basal (Tanner et al. Mol Cancer Ther. 2004 Dec;3(12):1585-92.)

<https://cellmodelpassports.sanger.ac.uk/passports/SIDM00963>

Name(s) BT-474

Tissue Breast

Cancer Type Breast Carcinoma

Tissue Status Tumour

Sample Site Unknown

Cancer Type Details Invasive Ductal Carcinoma not Otherwise Specified

Hormone receptor status and transcriptional subtype: ER+, PR+, Luminal B (Holliday and Speirs. Breast Cancer Research 2011, 13:215)

<https://cellmodelpassports.sanger.ac.uk/passports/SIDM00528>

Name(s) MDA-MB-361 (MDA-361)

Tissue Breast

Cancer Type Breast Carcinoma

Tissue Status Metastasis

Sample Site Brain

Cancer Type Details Breast Adenocarcinoma

Hormone receptor status and transcriptional subtype: ER+, PR-, Luminal B (Wang et al. Sci Rep. 2016; 6: 26456.)

However, we did analyse the lysosomes in the JIMT-1 cells as suggested and we do not detect any obvious lysosomal aggregation in the parental JIMT-1 cells. Furthermore, the lysosome morphology/aggregation is not altered upon SORLA-GFP expression.

Figure for reviewer: Staining of LAMP1 in GFP and SORLA-GFP expressing JIMT-1 cells.

7) The data on bladder cancer are interesting. However, in order to extend the mechanism proposed by the authors to bladder cancer cells (see Fig.5), authors should show that the KD of SORLA in 5637 cell model system causes HER2 intracellular accumulation and affects recycling, thereby reducing AKT signaling.

We appreciate that, ideally, it would be good to repeat all the functional assays with these cells as well. However, as this would be a substantial body of work, we chose to focus on some key experiments with these cells. First, we tried the suggested immunofluorescence experiment. However, since these cells are not HER2-amplified, but harbour an activating HER2 mutation, the levels of HER2 were so low (please see the HER2 cell-surface FACS data below for comparison of the cell lines, emphasizing this point) that with the antibodies we have available (with very good signal in the HER2-amplified lines) gave a very weak signal and did not allow us to reliably evaluate HER2 subcellular localisation.

Figure for reviewers: HER2 cell-surface levels in the indicated cell lines quantified by flow cytometry (n=5 from minimum of 3 independent experiments).

To overcome this, we analysed cell-surface levels of HER2 in control and SORLA-silenced 5637 cells. We find that SORLA silencing slightly reduces the already low cell-surface HER2 levels in these cells, although the result is not statistically significant (New Supplementary Fig. 6 and below for your convenience) indicating that also in these cells SORLA might regulate the cell surface residency of HER2.

Supplementary Fig. 6b: Cell surface labelling of HER2 in siCtrl and siSORLA#3 silenced cells (n=5 from 3 independent experiments). Unpaired t-test.

We also tested whether SORLA silencing sensitises the bladder cancer cells to CADs to extend the mechanism proposed in our manuscript to bladder cancer. We find that SORLA silencing increased the sensitivity of the bladder cancer cells to the antihistamine ebastine, similarly to the breast cancer cells (Fig 4g-i in the revised MS). These new data are included in the manuscript as Fig. 6f and below for convenience).

REVIEWERS' COMMENTS:

Reviewer #2 (Remarks to the Author):

The authors have now addressed all my queries very well.

Reviewer #3 (Remarks to the Author):

The revised version of the manuscript has significantly improved. Authors have performed additional experiments addressing all my major concerns. Clearly there is still variability in some of the observed responses, that highlight the difficulty of the HER2 system, in terms of tools and cell model systems available. I have only the suggestion to include in the text/legend a sentence highlighting the fact that although HER2 downregulation upon SORLA KD was observed consistently, its extent was variable among experiments.

The manuscript deals with a relevant issue in the fields of endocytosis and cancer biology, dissecting a novel mechanism of HER2 recycling mediated by SORLA, that appears to be relevant for tumor growth. I therefore recommend its publication.

Response to reviewers

Reviewer #1:

Reviewer #1 left only confidential comments to the editor but reiterated continuing concerns about the effect of SorLA on Her2 trafficking and turnover and ongoing lack of evidence for SORLA-HER2 interaction.

We therefore invite you to revise your paper one last time to address the remaining concerns of our reviewers. Notably, we will require toning down as appropriate of any SORLA-HER2 interactions as currently stated in the abstract and text. Please note in the absence of significant additions in the abstract, title, and text to address/note these potential concerns, we will be reluctant to move towards acceptance.

We apologise for the any unintentional lack of clarity regarding the SORLA-HER2 interaction. We do not claim a direct interaction between SORLA and HER2. We show by means of IP/pull-down that SORLA and HER2 can co-precipitate suggesting they may exist in the same molecular complex. We show that this complex is disrupted when the ECD of SORLA is lacking, suggesting that interactions at this domain are important for mediating a connection between SORLA and HER2. From the reviewer's comments, we now realize that, for example, the use of the word "association" may be misleading to someone reading the manuscript for the first time. We therefore thank the reviewer for pointing this out and we have removed any unclear reference to SORLA-HER2 association throughout the manuscript. We also include a sentence in the last paragraph of the introduction (and within the discussion) clearly stating that we do not know whether SORLA and HER2 interact directly or indirectly through an intermediary partner. We hope that these changes will alleviate the reviewer's concerns.

Reviewer #2:

The authors have now addressed all my queries very well.

We thank the reviewer for their valued input into the manuscript.

Reviewer #3

The revised version of the manuscript has significantly improved. Authors have performed additional experiments addressing all my major concerns. Clearly there is still variability in some of the observed responses, that highlight the difficulty of the HER2 system, in terms of tools and cell model systems available. I have only the suggestion to include in the text/legend a sentence highlighting the fact that although HER2 downregulation upon SORLA KD was observed consistently, its extent was variable among experiments.

The manuscript deals with a relevant issue in the fields of endocytosis and cancer biology, dissecting a novel mechanism of HER2 recycling mediated by SORLA, that appears to be relevant for tumor growth. I therefore recommend its publication.

We thank the reviewer for recognising the value of our paper to the field and for their continued insights into improving the manuscript. We have now added the following sentence (on page 6) to draw attention to the valid issue raised by this reviewer:

"Although the reduction in total HER2 protein levels upon SORLA silencing was observed consistently, its extent varied among experiments."